# Population-wide cerebellar growth models of children and adolescents

Carolin Gaiser [1,2], Rick van der Vliet[1,3,4], Augustijn A. A. de Boer[5,6], Opher Donchin[7,8], Pierre Berthet[9,10], Gabriel A. Devenyi [11,12], M. Mallar Chakravarty[11,12,13], Jörn Diedrichsen [14,15,16], Andre F. Marquand [5,6], Maarten A. Frens [1] ✉ & Ryan L. Muetzel[2,17]

In the past, the cerebellum has been best known for its crucial role in motor function. However, increasingly more findings highlight the importance of cerebellar contributions in cognitive functions and neurodevelopment. Using a total of 7240 neuroimaging scans from 4862 individuals, we describe and provide detailed, openly available models of cerebellar development in childhood and adolescence (age range: 6–17 years), an important time period for brain development and onset of neuropsychiatric disorders. Next to a traditionally used anatomical parcellation of the cerebellum, we generated growth models based on a recently proposed functional parcellation. In both, we find an anterior-posterior growth gradient mirroring the age-related improvements of underlying behavior and function, which is analogous to cerebral maturation patterns and offers evidence for directly related cerebello-cortical developmental trajectories. Finally, we illustrate how the current approach can be used to detect cerebellar abnormalities in clinical samples.

The cerebellum is known to be engaged in a broad spectrum of functions. While its involvement in motor control is best documented, recent efforts have made clear that it is also involved in cognitive function. Given that the cerebellum is strongly interconnected with the cerebral cortex, with cerebellar functional subunits being involved in a wide array of motor and cognitive tasks[1,2], these recent findings come as no surprise. Yet, despite converging evidence on the importance of

the cerebellum for brain function, limited work has explored how the cerebellum develops through childhood and adolescence.

The cerebellum is one of the first structures in the brain to start cellular differentiation, with a rapid growth period in the third trimester of pregnancy and in the first postnatal year, but it is one of the last to complete maturity[3,4]. Given this protracted developmental time course, it is especially vulnerable to genetic and environmental

[1]Department of Neuroscience, Erasmus MC, University Medical Centre Rotterdam, Rotterdam, The Netherlands. [2]Department of Child and Adolescent Psychiatry/Psychology, Erasmus MC - Sophia Children's Hospital, University Medical Centre Rotterdam, Rotterdam, The Netherlands. [3]Department of Neurology, Erasmus MC, University Medical Centre Rotterdam, Rotterdam, The Netherlands. [4]Department of Clinical Genetics, Erasmus MC, University Medical Centre Rotterdam, Rotterdam, The Netherlands. [5]Donders Institute for Brain, Cognition and Behavior, Radboud University Nijmegen, Nijmegen, The Netherlands. [6]Department for Cognitive Neuroscience, Radboud University Medical Center Nijmegen, Nijmegen, The Netherlands. [7]Department of Biomedical Engineering, Ben-Gurion University of the Negev, Be'er Sheva, Israel. [8]Zlotowski Center for Neuroscience, Ben-Gurion University of the Negev, Be'er Sheva, Israel. [9]Department of Psychology, University of Oslo, Oslo, Norway. [10]Norwegian Center for Mental Disorders Research (NORMENT), University of Oslo, and Oslo University Hospital, Oslo, Norway. [11]Cerebral Imaging Centre, Douglas Research Centre, McGill University, Montreal, Canada. [12]Department of Psychiatry, McGill University, Montreal, Canada. [13]Department of Biomedical Engineering, McGill University, Montreal, Canada. [14]Western Institute of Neuroscience, Western University, London, Ontario, Canada. [15]Department of Statistical and Actuarial Sciences, Western University, London, Ontario, Canada. [16]Department of Computer Science, Western University, London, Ontario, Canada. [17]Department of Radiology and Nuclear Medicine, Erasmus MC, University Medical Centre Rotterdam, Rotterdam, The Netherlands. ✉e-mail: m.frens@erasmusmc.nl

stressors disrupting development[3,5]. It thus could be a key node in various neurodevelopmental disorders and has the potential to serve as a crucial biomarker.

In addition to overlooking its role in higher cognitive function in the past, challenges in in vivo imaging of the cerebellum have likely hampered the study of this important structure. The anatomical location of the cerebellum and its tightly folded cortex have made it a more challenging structure to image since the acquisition field of view and head coils are often optimized for imaging the cerebrum. Additionally, high resolution images are needed for precise segmentations as well as anatomical and functional mapping. The adoption of high magnetic field strengths of 3 T and beyond in tandem with the development of dedicated automated segmentation and parcellation tools[6–9] has made the analysis of cerebellar imaging data more accessible.

As childhood and adolescence represent a time of increased risk for psychiatric and developmental problems[10], it is crucial to improve our understanding of cerebellar development during this period. For this reason, robust and detailed reference models of neurodevelopmental trajectories are needed, which recently has become a thriving area of research[11,12]. Normative modeling of brain imaging data is particularly well suited to this task and provides an analysis framework that is able to model biological heterogeneity at the level of the individual while also accommodating site effects[13,14]. This framework allows for tracking the development of a given individual against expected centiles in variation of a reference model, without needing to assume that clinical populations are homogeneous, analogous to growth charts in pediatric medicine. In the context of psychopathology, this approach has recently shown to increase sensitivity and to better characterize inter-individual heterogeneity in regional brain volumes compared to case-control studies[15,16]. By embedding normative modeling of imaging data within a federated learning framework, sharing of such models becomes possible without data privacy concerns. This not only means that smaller datasets can benefit from informative hyperpriors of a reference model based on much larger datasets, but also that models can be adapted and updated as more data becomes available[14]. Using this framework to establish normative models of the cerebellum will therefore prove particularly useful to detect deviations in cerebellar development on the level of individuals and to map these deviations to behavioral and clinical phenotypes.

Traditionally, the cerebellum has been subdivided in the medial-to-lateral direction into vermis and hemispheres, and in the anterior-to-posterior direction into lobules. However, more recently, *functional magnetic resonance imaging* (fMRI) has shown that functional boundaries in the cerebellum do not align with classical anatomical subdivisions[17]. Instead, an alternative functional parcellation containing at least 10 regions has been identified, which corresponds well to earlier proposed cerebro-cerebellar network parcellations[1], and that are characterized by the motor and cognitive features that elicit activity in the parcels[17].

In the current study, we describe and provide openly available normative models of anatomical and functional subregions of the cerebellum from a large pediatric population that (1) can be used as reference models to obtain accurate normative ranges, also in smaller datasets, by benefitting from informative hyperpriors based on a large sample, and that (2) can be updated with data from new sites and extended age ranges without the necessity of sharing sensitive patient or participant data. We furthermore illustrate the usefulness and practicality of the current approach by mapping the deviations from typical cerebellar development at the level of the individual in a sub-population of children with autistic traits[18]. These models have the potential to facilitate and maximize the use of cerebellar outcomes in neuroimaging research and as a result aid to better understand the role of the cerebellum in typical as well as atypical neurodevelopment.

## Results

### Sample characteristics and non-response analysis
A total of 7270 structural neuroimaging scans, stemming from 4862 unique individuals (2392 males, 2470 females) from the Generation R study, were available for statistical analysis (see Methods: Participants for details and exclusion criteria). Figure 1 shows the age and scanner distribution of the current sample. For normative modeling, the cohort was split into training (50%) and test (50%) sets (see Methods: Normative models). An overview of the sample characteristics of the training and test sets is shown in Supplementary Table 1 and allocation of the subjects to training and test sets resulted in groups that are representative of the cohort as a whole, also when considered at the sub-group level of measurement assessment.

### Normative models of the cerebellum
To generate normative models for anatomical and functional subregions of the cerebellum, we made use of the PCNtoolkit[12,19]. The PCNtoolkit allows for 1) obtaining normative ranges[19], 2) modeling individual heterogeneity to uncover clinically significant deviations[13,15,19], 3) correcting for batch-effects, such as differences between scanners[14], and 4) data sharing of the models for reuse without disclosing sensitive patient or participant information[14]. The *Hierarchical Bayesian Regression* (HBR) approach implemented in PCNtoolkit solves these problems by using shared priors from which the site-specific parameters and hyperparameters can be learned, and by providing a framework in which generated hyperparameters from previous analyses can be made available to new sites without the original data (i.e., a federated approach). Previously, we have shown that the HBR approach can be used to perform meaningful inferences across longitudinal time points, even when subsequent MRI assessments are conducted on different scanners[20]. Using HBR, we estimated normative models of the cerebellum using both volumes from an anatomical parcellation and morphological indicators (i.e., grey and white matter densities as well as volumes) from a functional parcellation for each *region of interest* (ROI) separately. Next to age as a predictor, sex and MRI scanner were modeled as batch-effects (see Methods: Normative models for further details on modeling parameters). A graphical representation of the normative model and its parameters can be found in Supplementary Fig. 1. Linear but also b-spline models were generated. Both perform equally well in this age range based on *Leave-one-out cross-validation* (LOO) (Supplementary Table 2). Following the principle of parsimony, the linear models will be described, however b-spline models are made available as well, as they might offer more flexible modeling options in future applications. Posterior distributions of all model parameters and ROIs converged

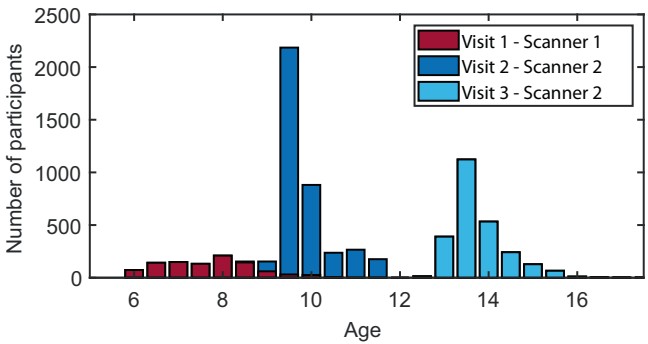

**Fig. 1 | Histogram of age and scanner distributions in the Generation R cohort.** Mean age visit 1: 7.9 years (range = [6.1–10.7], n = 974 [510 male, 464 female]), mean age visit 2: 10.1 years (range = [8.6–12.0], n = 3785 [1879 male, 1906 female]), and mean age visit 3: 14.0 years (range = [12.6–17.1], n = 2511 [1202 male, 1309 female]). 2734 (56.2%) individuals were measured once, 1848 (38.0%) individuals twice, and scans in all three measurement waves were acquired from 280 (5.8%) individuals. Source data are provided as a Source Data file.

well (>95% of $\hat{R}$ below 1.1[21]) and were visually inspected using a built-in function of the PCNtoolkit. In the following sections, ROIs of the anatomical and functional parcellations will be described separately.

## Anatomical parcellation

The cerebellum was parcellated in the native space into 35 anatomical subdivisions using the MAGeT pipeline[7,22]. In short, the MAGeT algorithm creates a multi-atlas based segmentation using a limited number of manually segmented atlases (here: five), which can be tailored to a specific cohort using a small set of study-specific template images. The resulting template-atlas library, allows for improved modeling of individual differences in morphology by taking advantage of the morphological variations in the representative template images (see Methods: Anatomical parcellation for details). Quality of segmentation was ensured by inspecting each scan visually (see Methods: Image quality control for details).

We fit a normative model to investigate age-related effects in volume for each of the 35 anatomical ROIs. Figure 2 illustrates the growth for each ROI using the mean posterior distribution of the age β coefficient (slope). Standardized coefficients are used to ease the comparison between outcomes, and results are stratified by sex. As expected for this age range, we see increasing volumes throughout all ROIs. The corpus medullare, the white matter of the cerebellum, shows the most marked increases in volume in both females and males. Interestingly, we see a growth gradient, starting with smaller age-related effects on volume in the anterior cerebellum (Lobules III – V), and increasingly larger age-related effects in the posterior cerebellum (Lobules VI – IX) with the largest effects, besides the corpus medullare, found in the flocculus (Lobules X). Growth trajectories of example ROIs in the anterior (left Lobule V) and posterior (left Crus I) cerebellum, as well as for the left corpus medullare are shown in Fig. 2B. Figure 2A also depicts sex differences in age β coefficients (slopes). No significant sex differences were observed, however, age-related coefficients were slightly higher for females (mean standardized β across ROIs = 0.178; 95% CI mean = [0.153 0.202]) than for males (mean standardized β across ROIs = 0.149; 95% CI mean = [0.128 0.170]), with larger effects in lobules VIIIA (left hemisphere: difference in standardized β = 0.142; and right hemisphere: difference in standardized β = 0.093), and left lobule X (difference in standardized β = 0.110). An overview of the age β coefficients for all ROIs can be found in Supplementary Table 3 as well as visually in Fig. 2C, and growth trajectories of all ROIs stratified by sex in Supplementary Figs. 2 and 3.

## Functional parcellation

As lobular boundaries of the cerebellum have shown limited correspondence with functional demarcations, we also employed the functional subregions proposed by King and colleagues[17] (see Methods: Functional parcellation for details). Ten functional regions of the cerebellum were identified using fMRI data from a large *multi-domain task battery* (MDTB) and labelled according to the cognitive and behavioral features that best described the task conditions (1: Left-hand (motor) presses, 2: Right-hand (motor) presses, 3: Saccades, 4: Action observation, 5: Divided attention (left hemisphere), 6: Divided attention (right hemisphere), 7: Narrative, 8: Word comprehension, 9: Verbal fluency, 10: Autobiographical recall). This parcellation was shown to successfully predict functional boundaries in a new set of motor, cognitive, affective, and social tasks, surpassing existing task-free and anatomical parcellations[17]. The 10 regions of the MDTB parcellation are illustrated in Fig. 3. Analogous to the anatomical parcellations described above, a normative model for the volume, *grey matter density* (GMD), and *white matter density* (WMD) was fit for each ROI of the functional parcellation. In Fig. 4A, we visualize the developmental trajectories of these functional parcels using again the mean posterior distribution of the standardized age β coefficient (slope).

Results are shown stratified by sex. Like previously seen in the anatomical parcellations, increases in volume are evident throughout all functional parcellations in males and females in this age range (Fig. 4A, a & d). Smaller age-related effects in volumes are present in the anterior parcels, known to be related to motor behavior, compared to posterior cerebellar regions, which comprise parcels associated with a range of cognitive processes. This trend can be seen even more strikingly in the GMD (Fig. 4A, b, e) and WMD (Fig. 4A, c & f) models. While it is well-documented that GMD decreases and WMD increases in the brain during this age range, we again see a clear distinction between anterior motor regions and posterior cognitive regions in 6 to 17 year olds. This is further illustrated by the growth trajectories of an example anterior motor (1: Left hand presses) and posterior cognitive (5: Divided attention (left)) ROI. Steeper slopes, and thus more developmental changes, are observed in posterior cognitive regions compared to anterior motor regions during childhood and adolescence (Fig. 4B a–c). Slight, albeit non-significant, differences in sex were observed with slower changes in GMD and WMD in females compared to males, particularly in the right hemisphere (mean standardized β across ROIs [95% CI mean]: volume males = 0.277 [0.251 0.304], volume females = 0.289 [0.256 0.322], GMD males = −0.138 [−0.206 −0.071], GMD females = −0.056 [−0.124 0.012], WMD males = 0.256 [0.165 0.347], WMD females = 0.184 [0.098 0.270]). Sex differences per ROI are illustrated in Fig. 4A g–i. As with the anatomical ROIs, an overview of the age β coefficients for all functional ROIs can be found in Supplementary Table 4 as well as visually in Fig. 4C and growth trajectories stratified by sex in Supplementary Figs. 4 and 5.

## Anterior-posterior growth gradient

Since both the anatomical and functional parcellation appeared to show smaller age-related effects in anterior compared to posterior parcels (Figs. 2, 4), we contrasted the growth trends in both parcellations by ranking the ROIs in terms of their anterior-to-posterior spatial position in the cerebellum. While the lobules in the anatomical parcellation are named in respect to their anterior-posterior placement (Lobule I to Lobules X), anterior-posterior positions for each functional parcel were obtained with a ranking procedure. Specifically, we ranked the functional parcels by determining the anatomical lobule in which their centroids (center point of each of the 10 functional ROIs) are located. In cases where the centroid was not located within lobular boundaries, but instead in regions of the corpus medullare, we visually assessed which lobule the centroid is closest to. In order to quantitatively assess any potential AP developmental gradient, we fit a linear trend line to the standardized age β coefficients (slopes) of all ROIs of the anatomical map (divided in vermal and mean of hemispheric ROIs) and functional map (divided in volume, GMD, and WMD) separately, as a function of anterior-to-posterior position (denoted as *AP growth coefficient* hereafter). Results are shown stratified by sex in Fig. 5A. An AP growth coefficient differing from zero suggest a statistically significant difference in growth of the cerebellum from anterior-to-posterior. We report uncorrected and the false discovery rate (FDR-BH) adjusted p-values of the AP growth coefficients.

In males, AP growth coefficients of the functional parcellation were significant in each modality ($\beta_{volume\_func} = 0.011$, $p_{volume\_func} = 0.019$, $pFDR_{volume\_func} = 0.032$; $\beta_{GMD} = -0.036$, $p_{GMD} < 0.001$, $pFDR_{GMD} = 0.002$; $\beta_{WMD} = 0.044$, $p_{WMD} = 0.003$, $pFDR_{WMD} = 0.010$) while the AP growth coefficients in the anatomical parcellations did not differ significantly from 0 ($\beta_{volume\_hemispheres} = 0.009$, $p_{volume\_hemispheres} = 0.060$, $pFDR_{volume\_hemispheres} = 0.075$; $\beta_{volume\_vermis} = -0.001$, $p_{volume\_vermis} = 0.569$, $pFDR_{volume\_vermis} = 0.569$). In females, the grey and white matter density AP growth coefficients of the functional parcellation as well as the hemispheric volume AP growth coefficients of the anatomical parcellation were significant ($\beta_{GMD} = -0.032$, $p_{GMD} = 0.005$, $pFDR_{GMD} = 0.010$; $\beta_{WMD} = 0.041$, $p_{WMD} = 0.003$, $pFDR_{WMD} = 0.010$;

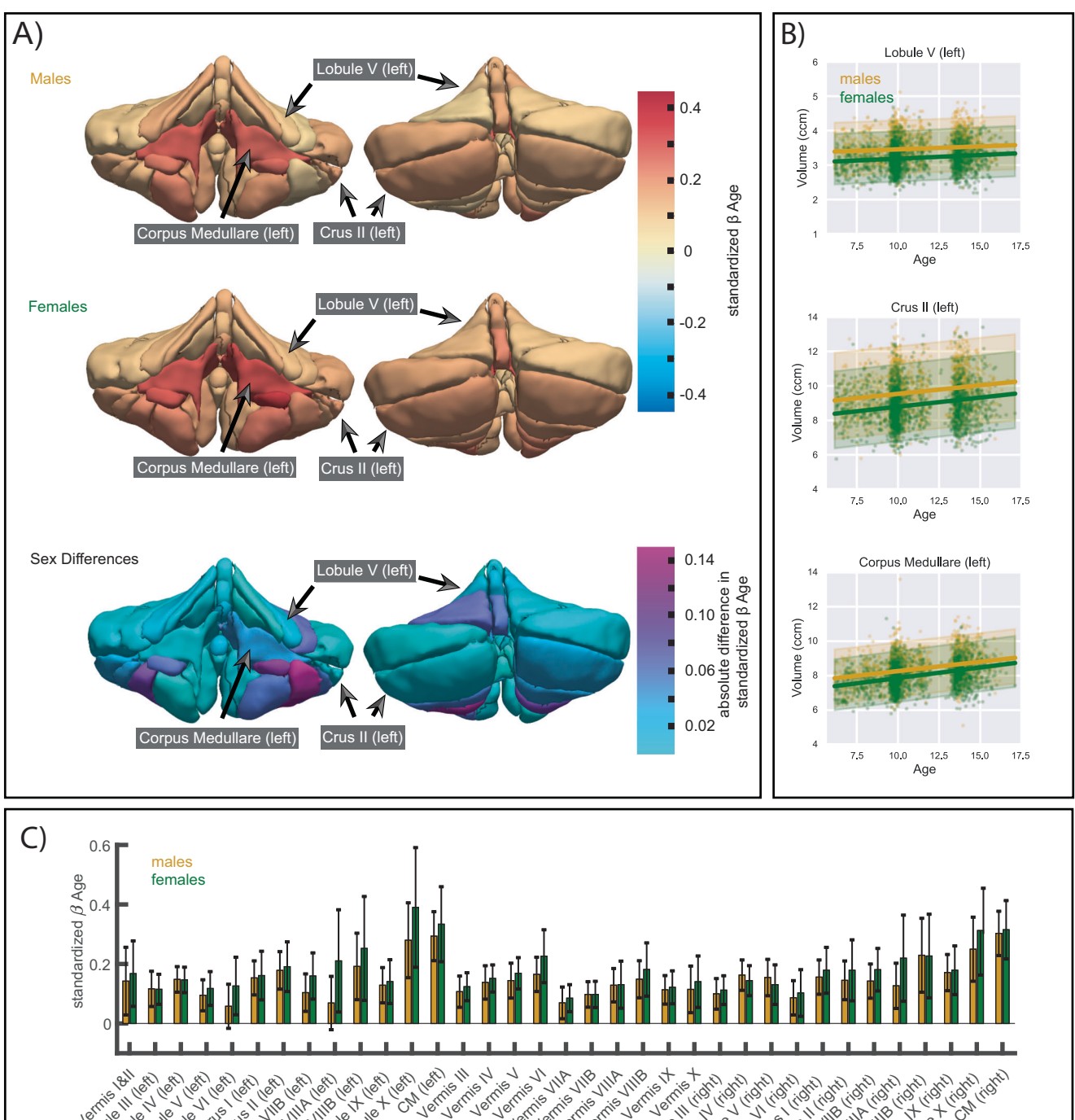

**Fig. 2 | Effect of age on volume in the anatomical parcellation. A** Mean posterior distribution for the standardized age β coefficient (slope) for each anatomical ROI and absolute differences in effect sizes (standardized β) between males and females are illustrated. 3D illustrations were generated based on a publically available, manually segmented MR image (MAGeT atlas, brain5)[7]. **B** Trajectories of males (in yellow) and females (in green) in 3 example ROIs: left Lobule V (anterior cerebellum), left Crus II (posterior cerebellum), and left corpus medullare (white matter tract). The bold lines represent the mean trajectories, shaded areas represent what is within 2 standard deviations of the mean. Volume is shown in cubic centimeters (ccm). **C** Bar graphs of all standardized age β coefficients (slopes) of males (in yellow) and females (in green). Error bars depict +/− 1 standard deviation of standardized age β samples ($n = 12{,}000$). Exact numbers can be found in Supplementary Table 3 and the percentage change of mean trajectories for each anatomical ROI is illustrated in Supplementary Fig. 6A. Source data are provided as a Source Data file.

$\beta_{volume\_hemispheres} = 0.016$, $p_{volume\_hemispheres} = 0.004$, $pFDR_{volume\_hemispheres} = 0.010$). The AP growth coefficient for volumetric changes in the functional parcellation did not survive multiple testing correction ($\beta_{volume\_func} = 0.013$, $p_{volume\_func} = 0.036$, $pFDR_{volume\_func} = 0.051$) and, as also seen in males, the AP growth coefficient of the anatomical vermis did not differ significantly from 0 ($\beta_{volume\_vermis} = -0.002$, $p_{volume\_vermis} = 0.544$, $pFDR_{volume\_vermis} = 0.569$).

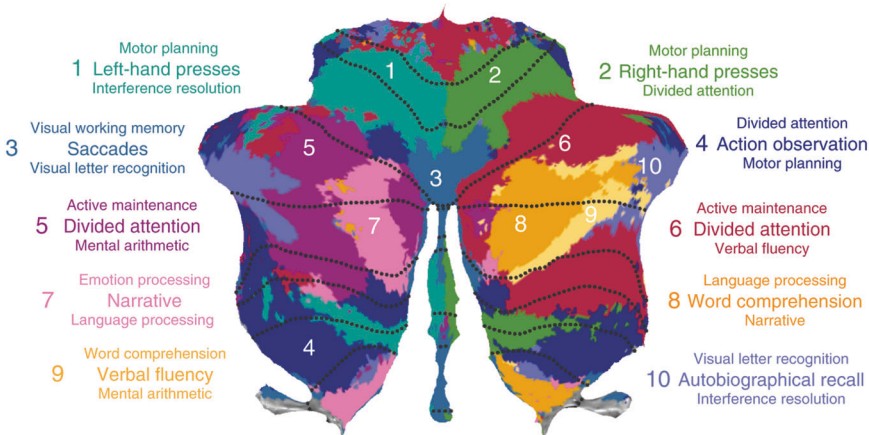

**Fig. 3 | Functional parcellation.** Figure adapted with permission from King et al. 2019[17] (https://www.nature.com/articles/s41593-019-0436-x). *Multi-Domain Task Battery* (MDTB) functional atlas regions are shown in color. Dotted black lines represent anatomical boundaries.

Lastly, we compared and visualized the gradients found using LittleBrain, a gradient-based tool to aid interpretation of topological neuroimaging findings of the cerebellum[23]. LittleBrain creates a two-dimensional representation of all voxels in the cerebellum, with each axis representing one of the principal functional gradients described by Guell and colleagues[24]. Gradient 1 stretches from primary motor to non-motor areas, such as language and default regions. This is analogous to functional organization principles previously reported in the cerebral cortex that extend from primary unimodal sensory to transmodal regions[25]. Gradient 2 can be understood as a characterization of task focus or cognitive load. This gradient ranges from the two extremes of gradient 1 (motor and default regions) to areas involved in focused cognitive processing, such as working memory or attention.

We mapped standardized age coefficient from the anatomical and functional parcellation using the LittleBrain toolbox and found that cerebellar growth during childhood and adolescence follows mostly gradient 2, and might thus be related to cognitive demands during development (Fig. 5B). Cerebellar regions with small age-related effects are mainly localized in task unfocused regions with low cognitive demand (low gradient 2 values), such as motor processing and default networks. Regions with larger age-effects can be found in task focused regions (high gradient 2 values), likely to overlap with frontoparietal networks[24]. Volumetric patterns, principally from the anatomical parcellation, show a more diffuse gradient pattern, possibly due to little overlap between functional activity and macroscopic anatomy[26].

## Large normative model deviations and clinical or behavioral phenotypes

To illustrate the utility of the cerebellar normative models, we examine whether deviations in cerebellar growth are present in children who are likely to fall on the Autism spectrum according to the *Social Responsiveness Scale* (SRS). The SRS has been shown to quantitatively assess subclinical and clinical autistic traits[18]. For each ROI, we contrast the z-scores between children likely to fall on the Autism spectrum (raw score on SRS >= 90th percentile, n = 198) to the remainder of the cohort with available SRS information (n = 2,012). The distribution of SRS scores can be found in Supplementary Fig. 7. Previously, it has been shown that using normative deviation scores can uncover more precise case-control effects and characterize clinically relevant differences in morphology on an individual level[13,15]. We therefore illustrate the percentage of children with high SRS scores that have a large deviation from the normative range (z > 1.96 / z < −1.96, critical z for 95% confidence interval) per ROI (see Methods for details). Using this definition, we expect roughly 2.5% of a typically developing population to have a large negative or a large positive z-score, respectively in

essentially all ROIs of the anatomical and functional parcellations. Indeed, this is what we observe in typically developing participants (Fig. 6, Supplementary Fig. 8). However, for participants with autistic traits (high SRS), this was not the case as more individuals than expected were represented in the extreme ends of the distributions. Considering the heterogeneity in brain morphology within ASD, it is important to emphasize that variations in individual deviation patterns exist among children at risk for Autism. These variations can be explored at the level of individual subjects using normative models.

In the anatomical parcellation, a higher percentage of participants with high SRS scores presented with large negative z-scores (smaller volume than expected) throughout various ROIs (Fig. 6A), specifically in vermal and hemispheric regions of the anterior and superior posterior cerebellum (significant percentage with large deviations binomial test at p < 0.05: Crus I (left), VIIIB (left), vermal region III, Lobule VI (right)). Large positive deviations (larger volumes than expected) can be observed in participants with low SRS (significant percentage with large deviations binomial test at p < 0.05: Left Lobules VI and Crus I, vermal region VIIA, and right Lobule VIIIB). In participants with high SRS scores large positive deviations appear to be less prevalent overall but significant positive deviations can be observed in the left Lobule IV and vermal region IX (p < 0.05).

Analogous to the anatomical parcellation, we also observe smaller volumes than expected throughout almost all functional parcels in participants with high SRS scores (Fig. 6B). But interestingly, subtle distinctions are revealed in the functional parcellation. While an overall trend towards smaller volumes than expected is visible in participants with high SRS scores (significant percentage with large deviations binomial test at p < 0.05: 1) Left-hand presses and 6) Divided attention (right)), the functional parcels best characterized by the cognitive features 7) narrative and 8) word comprehension seem to be exempted from this. In the GMD maps of large deviations in high SRS participants, more negative deviations than expected (lower GMD) in the left anterior cerebellum and posterior cerebellum, specifically in the parcellation best characterized by cognitive feature 9) verbal fluency (p < 0.05), can be observed. Moreover, we observed more positive deviations than expected (higher GMD) in the anterior motor areas of the functional atlas 1) Left-hand presses (p < 0.05) and 2) Right-hand presses in high SRS participants, as well as significantly higher GMD than expected in the functional region 4) Action observation (p < 0.05) in typical participants (Fig. 6C). Less pronounced differences are apparent in WMD deviations (Supplementary Fig. 8).

When the effect of SRS scores on normative deviation scores in both the anatomical and the functional parcellation was tested using linear regression models, volumes in anatomical lobules Crus II

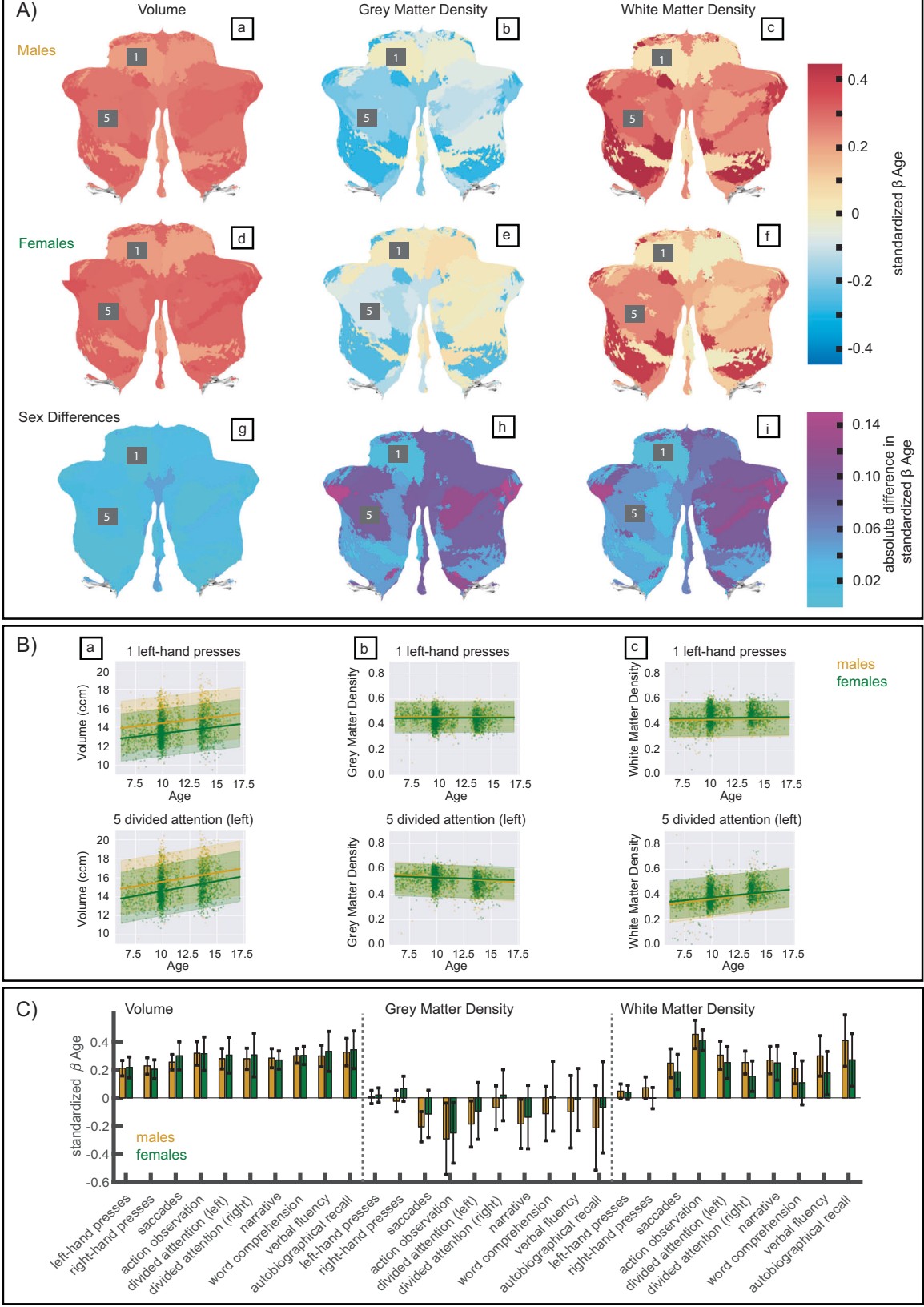

(hemispheric right), VIIB (hemispheric left and right), VIIIA (hemispheric left), and VIIIB (vermal) as well as functional parcels 4) action observation, 8) word comprehension and 10) autobiographical recall were associated with SRS scores. However, these associations did not survive multiple testing correction (Supplementary Table 6).

## Discussion

This study describes normative models of typical cerebellar development in a large pediatric population. Using over 7,000 longitudinal MRI scans, normative estimates of cerebellar growth from both anatomical and functional parcels were obtained by

**Fig. 4 | Effect of age on volume, *Grey Matter Density* (GMD), and *White Matter Density* (WMD) in the functional parcellation. A** Mean posterior distribution for the standardized age β coefficient (slope) for each functional ROI of the MDTB atlas (a-f) and absolute differences in effect size (standardized β) between males and females are illustrated (**g–i**). **B** Trajectories of males (in yellow) and females (in green) for 2 example ROIs. 1: Left hand presses (anterior cerebellum) and 5: Divided attention (left) (posterior cerebellum). The bold lines represent the mean

trajectories, shaded areas represent what is within 2 standard deviations of the mean. **C** Bar graphs of all standardized age β coefficients (slopes) of males (in yellow) and females (in green). Error bars depict +/− 1 standard deviation of standardized age β samples (n = 12,000). Exact numbers can be found in Supplementary Table 4 and the percentage change of mean trajectories for each functional ROI is illustrated in Supplementary Fig. 6B. Source data are provided as a Source Data file.

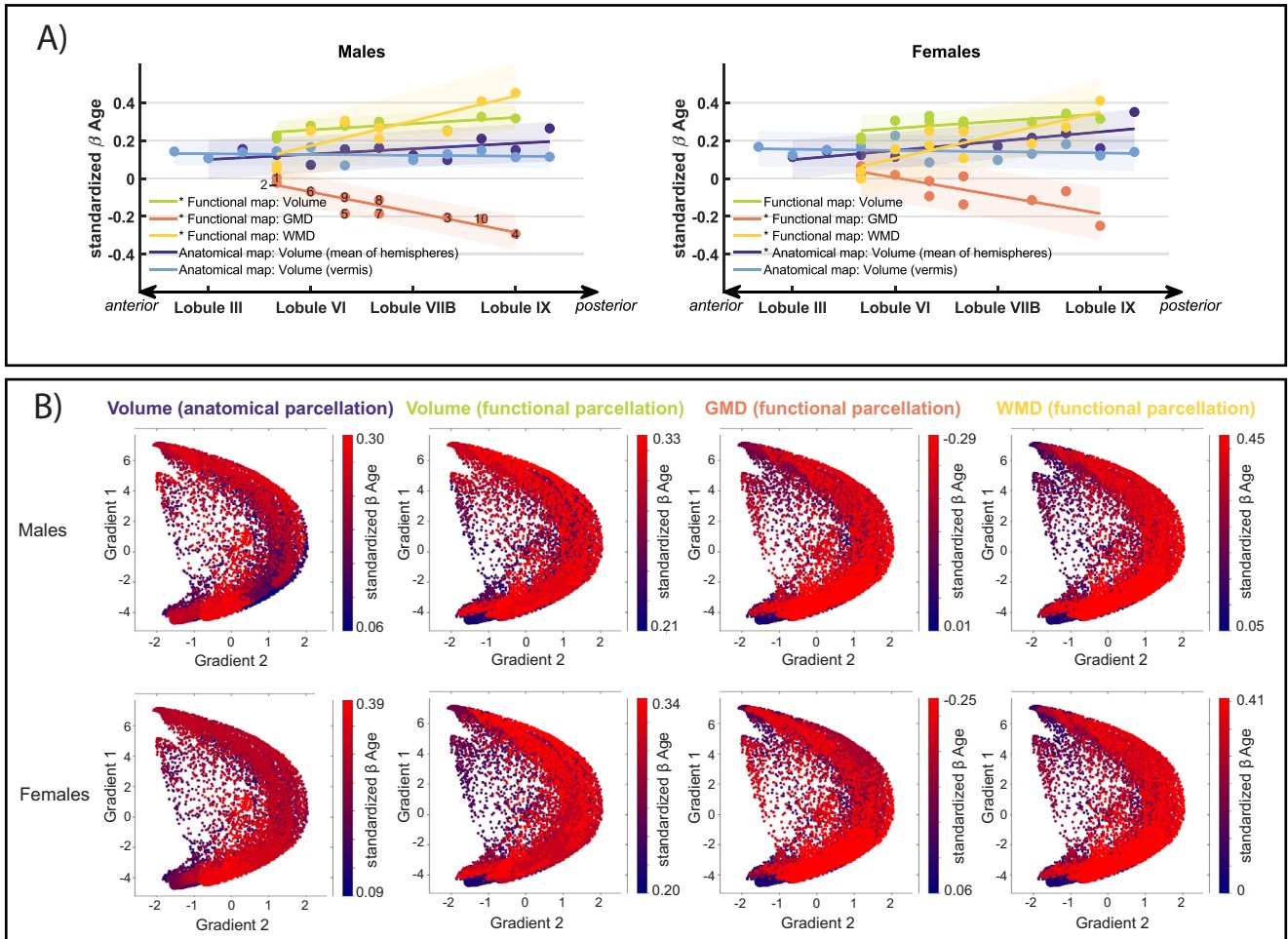

**Fig. 5 | Visualizations of growth gradients. A** Linear fit lines through the standardized age β coefficients (shown as dots) of anatomical (vermal in light blue and mean of both hemispheres in dark blue) and functional (volume in green, *Grey Matter Density* (GMD) in orange, and *White Matter Density* (WMD) in yellow) ROIs in anterior-to-posterior order. Shaded areas indicate the 95% prediction intervals of the linear fit lines. Asterisks in the legend indicate significant AP growth coefficients (slopes of linear fit lines). Anatomical location of functional parcellation centroids are indicated by numbers in the first panel and listed in Supplementary Table 5.

**B** Growth gradients visualized along two functional gradients using the LittleBrain tool[23]. Gradient 1 (y-axis) ranges from motor (negative values) to non-motor areas (positive values); Gradient 2 (x-axis) from low (negative values) to high (positive values) task focus/cognitive load. Each dot in the scatterplot represents a voxel in the cerebellum. The color map (scaled per modality to ease comparisons) shows standardized age β coefficients of the cerebellar parcellation a given voxel belongs to. Source data are provided as a Source Data file.

fitting hierarchical Bayesian regression models. Despite its potential to serve as a biomarker for developmental disorders[5,27,28], the morphology of the cerebellum has never been studied at this scale and detail using human MRI data before. Previous projects on normative modeling have demonstrated the potential impact and demand for large, open-source growth models[11,13]. However, presently available models fail to include the cerebellum. The current models will therefore be critical to understand the heterogeneity in cerebellar development as well as deviations from the normative range in disorders. Notably, these deviations can be investigated on the level of a single individual using the current approach.

The anatomical (MAGeT atlas[7]) and functional regions (MDTB atlas[17]) show similar overall growth trends. As expected in this age range traversing late-childhood into adolescence, we see increasing volumes throughout all ROIs in both parcellations (Fig. 2A, Fig. 4A a, d). Consistent with previous findings, we observed an anterior-posterior gradient in cerebellar development likely to reflect and mirror the age-related improvements in underlying functions, with sensorimotor areas predominantly located anteriorly and cognitive areas posteriorly in the cerebellum[17,29,30]. Between ages 6 and 17, anterior sensorimotor areas show smaller age-related effects compared to posterior cognitive areas, possibly reflecting protracted growth trajectories for higher-order cognitive compared to sensorimotor regions in the cerebellum.

**A)** Anatomical parcellation: Volume

**B)** Functional parcellation: Volume

**C)** Functional parcellation: GMD

While, upon visual inspection, a growth gradient from anterior to posterior can be observed in both parcellations, testing this trend quantitatively is challenging. We tested the growth trend in both parcellations and in different morphological indicators by ranking the standardized age coefficients (slopes) per ROIs in terms of their anterior-to-posterior position and determined a linear fit line (Fig. 5A).

Resulting slopes of these fit lines (*AP growth coefficients*) were significant, and thereby add additional support to the rationale of a growth gradient, in the functional parcellation (volumes, GMD, and WMD) for males and in the functional (GMD and WMD) as well as anatomical (mean volume of hemispheres) parcellation for females. AP growth coefficients of vermal volumes in the anatomical parcellation

**Fig. 6 | Cerebellar deviations of typically developing children and children with autism traits.** Percentage of individuals with large negative (z-score < −1.96) and large positive (z-score > 1.96) deviations in typically developing children (n = 2,012) and children that are likely to fall on the Autism spectrum (high *Social Responsiveness Scale* (SRS) score, n = 198) are shown. Asterisks indicate ROIs in which children with high SRS and children with typical SRS scores have a significantly higher percentage of large deviation than expected at the p = 0.05 level (typical > 3.13%, high SRS > 5.05%) using Binomial testing (observed vs. expected number of participants with z > 1.96/z < −1.96 in high SRS and typical children, given a null hypothesized probability of p0 = 0.025, one-sided). **A** Deviations in volume in the anatomical ROIs. 3D illustrations were generated based on a publically available, manually segmented MR image (MAGeT atlas, brain5)[7]. **B** Deviations in volume in functional ROIs. **C** Deviations in *Grey Matter Density* (GMD) in functional ROIs. *White Matter Density* (WMD) deviations are shown in Supplementary Fig. 8. Source data are provided as a Source Data file.

were not significant. The possible absence of an anterior-posterior growth gradient in vermal areas is interesting, since functional differences are also present throughout the vermis. However, these are functionally distinct from the hemispheres of the cerebellum as the vermis receives sensorimotor afferents from the spinal cord, and is predominantly involved in lower-order functions, such as postural control, locomotion, and gaze[31], but also plays an important role in emotion processing[32–34].

Nevertheless, a note of caution is due here since ranking the ROIs in terms of position is not unambiguous. This is a result of the cerebellar geometry; the anterior-posterior axis of the cerebellum resembles a C-shape, rather than a line or a plane, making the ranking more intricate. While anatomical lobules follow an anterior-posterior order, correcting for the inter-lobular spacing along this anterior-posterior axis is equivocal due to differences in lobule sizes. The position of the functional ROIs on this anterior-posterior axis becomes even more difficult to distinguish, since functional parcels are not contiguous and might have anterior and posterior components (e.g. parcels 1: Left-hand (motor) presses, and 2: Right-hand (motor) presses). By determining the centroid for each of the 10 functional ROIs, we approximated the anterior-posterior position. Using this approach, we present evidence for an anterior-posterior gradient throughout different morphological indicators (i.e., volumes as well as GMD, and WMD) that is observable in both parcellations with the exception of vermal volumes in the anatomical parcellation. However, results should be interpreted with caution given the complex geometry of the cerebellum and the possibility of non-linear functional gradients[1].

Interestingly, the anterior-posterior growth trends in the cerebellum mirror a previously reported cerebellar functional gradient and maturation patterns found in the cerebrum. In the cerebral cortex, similar patterns of earlier maturation of sensorimotor compared to higher-order cognitive areas can be observed in myelination[35,36] and grey matter maturation[37–40], pointing towards directly related growth trajectories of the cerebellum and the cerebrum. Considering the two principle functional gradients that have been described by Guell and colleagues, our reported growth gradients might not only be reflective of motor vs. non-motor involvement but might also pertain to cognitive load. The first gradient extends from motor to non-motor areas, while the second gradient stretches from areas involved in task focused to task unfocused processing[24]. The growth gradients in both parcellations and over different modalities (volume, GMD, WMD) map best onto gradient two, although volumetric gradient patterns seem to be more diffuse (Fig. 5B). This implies earlier maturation of areas involved in task unfocused cognitive processing, like motor function and default mode networks, and later or prolonged maturation of regions likely to share involvement in working memory processing and frontoparietal networks. In the cerebrum, frontoparietal networks are known to mature later as, for example, sensorimotor networks. However, recently age-dependent maturation patterns paralleling the first, but not the second functional cerebellar gradient have been reported, with default networks reaching maturation last in a very similarly aged cohort[41]. An alternative explanation for this result could have to do with the interplay between default mode and frontoparietal networks. The default mode network has been proposed to serve as a compensatory scaffold to support executive functions in children and young adults with immature frontoparietal network[42]. Importantly, while there are suggestions of the growth patterns resembling gradient 2 proposed by Guell and colleagues, the depiction of the gradients using the LittleBrain toolbox in the current study is inconclusive, mainly due to divergent patterns in the anatomical parcellations. The issue of age-related changes in cerebellar functional networks remains to be closely examined and could be explored in future studies using large-scale, longitudinal functional neuroimaging data. Together with similar maturation patterns in myelination and grey matter density, these findings provide additional support for developmental interactions between the cerebellum and cerebrum.

Indeed, the cerebellum has been proposed as a crucial node for optimal structural and functional brain development. Hence, Wang and colleagues have recently coined the term of a *developmental diaschisis*, suggesting that the cerebellum might have a direct influence on cortical maturation[5]. This also accords with earlier findings of volume decreases in remote but connected cerebral regions after perinatal cerebellar injury[43] and findings of cerebellar tumors resulting in significant downstream effects on higher cognition and motor function which could not be compensated well by other structures[44]. Further work, in particular in vivo human research, is required to develop a more complete understanding of the cerebellar influence on cortical development.

An increasing amount of literature about the role of the cerebellum in higher cognitive functions but also in neurodevelopmental disorders, and in *Autism Spectrum Disorder* (ASD) specifically, has become available in recent years. Cerebellar abnormalities are among the most frequently reported in ASD patients[5]. Intriguing reports from mouse models show that targeted activation of right Crus I and the posterior vermis was able to rescue autistic behaviors in TSC1 mutant mice by modulating activity in the medial prefrontal cortex[45]. Akin volumetric changes in Crus I and the posterior vermis have been reported in human studies as well as deviations in total cerebellar size[46–51]. Recently, however, no differences in cerebellar anatomy in individuals with autism were reported when using normative models on cerebellar growth based on a smaller control sample (n = 219)[52]. Therefore, looking into cerebellar deviations of children that are likely to fall on the Autism spectrum in large, population-based cohorts, lends itself as a prime example of illustrating the utility of the current normative models.

In accordance with previous research, we find smaller cerebellar volumes in children with autistic traits. In the anatomical parcellation smaller volumes can be seen throughout various regions, particularly in vermal and lobular parts of the anterior and superior posterior cerebellum (Fig. 6A). This corroborates findings of hypoplasia in posterior vermal and lobular regions, which have been reported consistently before in clinical samples[46–48,50,51]. While the functional parcellation also reveals smaller volumes throughout almost all ROIs, with significant differences in MDTB components 1) Left-hand presses and 6) Divided attention (right), the MDTB components 7) Narrative and 8) Word comprehension seem to not follow the same trend (Fig. 6B). Given the overlap of MDTB components 7 and 8 with the cerebellar default mode regions described by Buckner and colleagues[1], a network found to be among the most disrupted in ASD patients, this might relate to previously reported heterogeneity in default mode network connectivity in children on the Autism spectrum[53].

Clear differences can also be observed in GMD with a high percentage of individuals with autistic traits exhibiting increased GMD in the anterior sensorimotor parcels, particularly on the left hemisphere, and decreased GMD in the superior posterior parcels involved in language processing (Fig. 6C). In view of the heterogeneity in brain morphology between individuals across a multitude of pathologies, it is noteworthy that the current approach does not depend on group-level inferences but can be used at an individual level to uncover within-group differences. Furthermore, the availability of a very detailed anatomical as well as a functional parcellation allow for more sensitive approaches when investigating pathological deviations from typical development.

While we chose to illustrate cerebellar deviations using the example of autistic traits, it is important to note that the cerebellum is known to play an influential role in a myriad of clinical subpopulations for which this approach would be particularly insightful. Following the concept of the cerebellar connectome, a framework proposing that deviations in cerebellar and cerebello-cortical connectivity have a direct influence on onset and severity of neurodevelopmental disorders, the current approach has the potential to not only advance our understanding of disease etiology, but might also uncover new sites for therapeutic interventions[28].

Previous investigations into the typical volumetric development of the cerebellum during childhood and adolescence, although conducted with smaller sample sizes and employing less detailed anatomical parcellations, have revealed similar growth patterns. Increases were most pronounced in the corpus medullare and superior posterior lobe, while the anterior lobe and vermal regions showed stagnation or decline[54,55]. Contrasting anatomical and functional maps in the current study suggests there is merit in using both and that in certain applications they may be complimentary. Anatomical maps offer a well-defined way of delineating the cerebellum, thereby providing parcellations that have shown to successfully quantify brain morphology and pool data across populations in the past. But as functional activity in the brain rarely coincides with macroscopic anatomy[26], future studies employing functional parcellations might be able to uncover predictors of behavior in clinical subgroups that would otherwise be hidden using anatomical atlases only.

Therefore, normative models of both parcellations and all ROIs described are freely available on github (https://github.com/cgaiser1/cerebellar-growth-models)[56]. Both linear as well as b-spline models can be downloaded and used as informative priors for new unseen sites using the PCNtoolkit (https://github.com/amarquand/PCNtoolkit.git). This not only allows for better predictions in smaller data sets, but also enables future studies to model individual differences free of site-effects, and to uncover clinically significant deviations from the normative model. To transfer knowledge from the current models to a new cohort an adaptation set of approximately 25 samples is needed[20]. The remainder of the cohort can then be interpreted on a single subject basis and compared with the reference model without the need to employ an additional control group. A detailed account of how normative models can be implemented in future studies can be found online (https://pcntoolkit.readthedocs.io/en/latest/) and is also described by Gaiser, Berthet and colleagues[20]. Furthermore, an approach to quantitatively evaluate within-subject changes in longitudinal designs using normative modeling has been proposed recently[57], which promises great utility for individual-level data as it allows to estimate whether individual participants or patients follow their expected centiles. The PCNportal (https://pcnportal.dccn.nl/), where the current models can be found as well, further offers the possibility to derive subject-level statistics in a new dataset in a simple and accessible way, without the need for any technical background knowledge or computing power[58]. Greater efforts are needed to discern the role of the cerebellum in typical and atypical neurodevelopment for which the current normative models can serve as a highly useful tool.

## Limitations

The Generation R Study is a population-wide cohort from The Netherlands and therefore the current normative models might not generalize ideally to other populations. Participants from Western, educated, industrialized, rich, and democratic (WEIRD) societies are often over-represented in research samples, particularly in small studies[59]. Yet, importantly the Generation R study is a population-based multi-ethnic cohort[60] and in the current study sample 29.8% of participants come from non-European backgrounds (Supplementary Table 1). Also, the distribution of IQ scores in the current sample closely follows the population distribution (mean $\pm$ STD = 102.3 $\pm$ 14.8). While normative models were found to be highly stable with $n > 3,000$[61], it is noteworthy that the current models can easily be updated within the PCNtoolkit framework. This can also include new data points outside of our age range and from diverse backgrounds, or from clinical cohorts. As a consequence, the cerebellar normative models can be extended and refined as more information becomes available while new, possibly smaller cohorts can benefit from informed priors based on our models. In such a way, the current models and our results on deviations in children likely to fall on the autism spectrum can be validated in an external (clinical) cohort in the future.

Another potential limitation is related to how the MRI data were processed, namely the use of an adult template space and adult atlases. This is a recurring theme in human neuroimaging, which has yet to be fully resolved. Importantly, the MAGeTBrain framework is likely able to improve modeling of individual differences in morphology, possibly present in a developing cohort compared to an adult population, through the use of study-specific template propagation. Furthermore, our scans were acquired on 3 T MRI scanners with a voxel resolution of 1 mm$^3$. While this standard setup is able to reliably identify cerebellar lobules[7], higher resolutions are needed to more accurately segment cerebellar vermal regions, and white and grey matter given the thin, tightly folded cortical layering of the cerebellar cortex[62]. Lastly, the functional parcellation used in this study was a group-average parcellation derived from high-quality, extensive functional MRI assessment from adults[17]. Functional boundaries are likely to vary between individuals to some degree and may also vary as a function of age. Future studies should therefore aim to repeat the task-battery in a cohort of young children and adolescents to quantify whether neurodevelopmental differences exist in the functional parcellation.

In conclusion, we present models of cerebellar growth during childhood and adolescence, an important time period for brain development, based on a large, prospective population cohort, the Generation R study. We find an anterior-posterior growth gradient mirroring the age-related improvements of underlying behavior and function. The anterior/sensorimotor-posterior/cognitive growth gradient resembles a recently proposed functional gradient related to cognitive load and follows cerebral maturation patterns, thus providing evidence for directly related cerebello-cortical developmental trajectories. In recent years, the cerebellum has received increasing attention as a critical node in fundamental cognitive and emotional functions as well as brain development. The current openly accessible growth models will therefore be of great value for uncovering cerebellar deviations and understanding their implications in neuropathology.

## Methods
### Participants

Participants were part of the Generation R Study, a population-based, prospective cohort study from fetal life onward[60]. Between 2002 and 2006, 9,778 pregnant women living in Rotterdam, The Netherlands were enrolled in the study. Data from the children and caregivers were collected at several time points. In total, MRI data from 5,185 unique individuals were obtained across three time points. 1,070 participants (mean age = 7.9) visited during the first assessment, 3,992 participants

(mean age = 10.2) during the second, and 3,725 participants (mean age = 14.0) visited the testing center at the third assessment. After exclusion of participants with incomplete $T_1$-weighted scans ($n = 1,214$), scans without complete consent form ($n = 122$), scans with incidental findings ($n = 73$), and scans with low image quality ratings ($n = 454$), a total of 7,270 scans from 4,862 individuals (2392 male, 2470 female) were available for statistical analysis. Scans were only excluded based on technical or ethical considerations, but not based on clinical phenotypes (i.e. pre-existing conditions) in order to capture the heterogeneity of the general population. In the first measurement assessment, participants included in the analysis had a mean age of 7.9 years (range = [6.1-10.7], $n = 974$, 510 male, 464 female), in the second assessment a mean age of 10.1 years (range = [8.6 − 12.0], $n = 3,785$, 1,879 male, 1,906 female), and in the third assessment a mean age of 14.0 years (range = [12.6 − 17.1], $n = 2,511$, 1,202 male, 1,309 female). 2,734 (56.2%) individuals were measured once, 1,848 (38.0%) individuals twice, and scans in all three measurement assessments were acquired from 280 (5.8%) individuals. Roughly half of the sample was female (50,6%), 29.8% were of non-European ancestry and IQ scores of included participants closely followed the population distribution (IQ mean ± STD = 102 ± 14.8) (Supplementary Table 1). Written informed consent from both parents and assent from all participants was obtained, and the study was approved by the Medical Ethical Committee of the Erasmus Medical Center. Participants did not receive monetary compensation, but their travel costs were reimbursed. Additionally, as a token of appreciation for their participation, they received small gifts valued at 10€ or less, such as a drinking bottle, a bag, a power bank, or similar items.

## Non-response analysis

Given the prospective, longitudinal nature of the study, it is important to understand the impact of loss-to-follow-up. We therefore tested random drop-out in our study by examining possible differences between participants included and excluded in the current analysis in terms of the following descriptive characteristics: sex, parental national origin (Dutch, Non-Dutch but European, Non-European; obtained from birth records), monthly household net income (low = <1200€, middle = 1200€ − 3200€, high = >3200€; obtained from questionnaire), maternal education (higher education pursued or not; obtained from questionnaire), IQ and behavioral problems. Nonverbal IQ scores, normalized for sex and age, were measured using 2 subtests (Mosaics [spatial visualization] and Categories [abstract reasoning]) of the *Snijders-Oomen Nonverbal Intelligence Test* (SON-IQ) in the first measurement visit (mean age = 7.9). Behavioral problems were measured using the *Child Behavioral Checklist* (CBCL)[63] in the second measurement visit (mean age = 10.1). We dichotomized behavioral problems according to maternal reports (scoring above 80th percentile: behavioral problems present; below 80th percentile: behavioral problems not present).

Participants included in the dataset were more likely to be female ($\chi^2 = 8.38$, $p = 0.004$), of Dutch national origin ($\chi^2 = 181.10$, $p < 0.001$), have higher maternal education ($\chi^2 = 27.10$, $p < 0.001$), and to have a middle or high household income ($\chi^2 = 15.48$, $p < 0.001$) and less likely to have an IQ below 85 ($\chi^2 = 63.21$, $p < 0.001$). Behavioral problems as measured by the CBCL did not differ between excluded and included participants (weighted total problem score; $\chi^2 = 2.41$, $p = 0.120$).

## Neuroimaging acquisition

Scans were acquired using two different MRI scanners. In the first measurement visit, data were collected on a GE MR750 Discovery system, and data from all subsequent assessments were collected on a study-dedicated GE MR750w system (General Electric Healthcare, Wisconsin, USA). High resolution $T_1$- weighted MRI scans were acquired using an inversion recovery fast spoiled gradient recalled sequence (IR-FSPGR) using the following parameters: Visit 1: $T_R = 10.3$ ms, $T_E = 4.2$ ms, $T_1 = 350$ ms, flip angle = 16°, acquisition time = 5 min 40 s, field of view = 230.4 × 230.4 mm, 0.9 × 0.9 × 0.9 mm$^3$ isotropic resolution. Visit 2 and 3: $T_R = 8.77$ ms, $T_E = 3.4$ ms, $T_1 = 600$ ms, flip angle = 10°, acquisition time = 5 min 20 s, field of view = 220 ×220 mm, 1x1x1 mm$^3$ isotropic resolution.

## Image pre-processing

Images from the first measurement visit were resampled to 1 mm isotropic resolution to match data from the second and third assessments. Images were then pre-processed using the SMRIPrep tool[64]. Briefly, non-brain tissue was removed, voxel intensities were adjusted for $B_1$ inhomogeneities, and images were then linearly and eventually nonlinearly registered to standard stereotactic space (MNI152 NLin2009cAsym 1x1x1 mm resolution) using ANTs (github: https://github.com/ANTsX/ANTs.git). The tissue segmentation procedure (FSL FAST: https://fsl.fmrib.ox.ac.uk/fsl/fslwiki/FAST) resulted not only in binary classifications of voxels, but also in per-voxel tissue class probability estimates. These probability estimates can be interpreted as the likelihood of a given voxel being grey matter, white matter, or cerebrospinal fluid. Further, the nonlinear registration produced a nonlinear warp file (which included the linear initialization) from which we calculated the determinant of the Jacobian matrix for each voxel. This determinant was used as a measure of volume of that voxel relative to its volume in standard stereotactic space.

## Normative models

Normative models for anatomical and functional subregions of the cerebellum were generated using the PCNtoolkit python package version 0.27[12,19,65] using Python 3.10.6. We modelled the effect of age on cerebellar features of interest (volumes, GMD, WMD) while correcting for batch-effects of sex and scanner (model parameters are illustrated in Supplementary Fig. 1). For both the anatomical and functional parcellations, we split the dataset into a training set (50%) and test set (50%) using the sex and scanner site variables to ensure equal distribution of sex and both scanners in both sets. We generated linear and 3rd order b-spline models with 5 evenly spaced knot points (all available at: https://github.com/cgaiser1/cerebellar-growth-models). Normative models of all previously described cerebellar ROIs were successfully generated. Model performance of both linear and b-spline models were evaluated using *Leave-one-out cross-validation* (LOO). We employed a sinh-arcsinh likelihood (SHASHb) to accommodate non-Gaussian distributions[65] and modeled random effects in intercept, slope, and variance (sigma) on the batch-effects (sex and scanner). Inference was performed using Markov chain Monte Carlo methods (see Kia et al., 2022 and de Boer et al., 2022 for full details). Four chains with 2000 samples each were generated. The first 500 samples of each chain were used as tuning samples and were removed from further analysis. Model outputs include the posterior distributions of the parameters and deviations from the normative range (z-score which are free of batch-effects) for each individual in the test set.

## Image quality control

To ensure segmentation quality, anatomical segmentations ($n = 8787$ before exclusions) were visually inspected by two expert raters (C.G., N.D.). A custom-made MATLAB app (version R2021b, Mathworks, USA) was used to inspect PNG files of all slices of each scan, and the segmentation quality was rated on a 3-point scale (Good, Sufficient, or Bad) based on inaccuracies in the parcellation, complete coverage of the cerebellum, and motion or other artifacts. Scans rated as bad (i.e., cases without full coverage of the cerebellum, scans with substantial artifacts, and/or scans with marked inaccuracies in the parcellation) were subsequently excluded from further analyses. Visual examples of scans rated on the 3-point scale can be found in Supplementary Fig. 9. A subset of 600 scans were inspected by both raters to assess *inter-rater reliability* (IRR). Following visual inspection of all available scans, 454

(5.2%) were excluded due to low quality ratings, 593 were rated as sufficient (6.7%), and 7740 as good (88.1%). IRR between both raters (C.G., N.D.) showed strong agreement on usability (usable or not usable) of the scans between raters (Cohen's $\kappa$ = 0.83, CI = [0.72–0.94]).

### Anatomical parcellation
The MAGeTBrain framework uses an automated labeling algorithm based on five manually segmented MR images from healthy participants. Non-linear registration is used to align the five manually segmented images (in MAGeTBrain referred to as "atlases") to a series of individual study-specific "template" images (referred to as "templates" here).

Seven unique and representative images from the three study time points were selected, by (a) excluding scans with dental implants, (b) pre-selecting the top 20 scans with the highest quality ratings based on automatic FreeSurfer quality assurance[66], and (c) thoroughly inspecting the top 20 scans per time point for artifacts, inhomogeneities, full coverage of the cerebellum, and (cerebellar) cysts. Based on this evaluation, the 7 scans with the highest quality ratings were selected per time point, resulting in 21 study-specific template images.

Each of the five manually segmented atlases were then applied to the 21 study-specific templates, resulting in 105 cerebellar atlas-template segmentations. This allows for the manual segmented atlases to be propagated to each of the template images. Next, each individual scan in the dataset was non-linearly registered to each of the 105 cerebellar template segmentations, resulting in 105 segmentations for each input image, and enabling the template-atlas labels to be propagated to each individual participant space[22]. In the final step, for each individual input image, the 105 co-registered atlas-template labels were then fused using voxel-wise majority voting to create a final segmentation.

The MAGeT algorithm subdivides the cerebellum into 11 vermal and 22 hemispheric lobules (11 on each hemisphere). Additionally, the central white matter, the corpus medullare, is segmented in each hemisphere. White matter that extends into the folia of the lobules was segmented as part of the lobules. Volumes for each of these 35 anatomical parcellations are generated in mm³ by the MAGeT pipeline. Supplementary Fig. 10 shows a representative automatically labelled segmentation from one individual and gives an overview of all 35 anatomical subdivisions. This computationally intensive approach has been shown to have better test-retest reliability than other segmentation techniques and results in high segmentation accuracy by creating a customizable template segmentation library, thereby being able to take advantage of existing morphological variances in the cohort[7].

### Functional parcellation
We parcellated functional subregions of the cerebellum in MNI space, using the MNI-aligned version of the MDTB atlas[17]. The MNI-aligned atlas was chosen over the SUIT space aligned atlas, since it allows to use the current models without adding an additional processing step transforming images into SUIT space. Furthermore, while normalization to SUIT space compared to the former linear MNI template has been shown to improve overlap of cerebellar regions across individuals[67], alignment to the MNI152NLin (the non-linear MNI template published since) has greatly improved MNI normalization since then. Mean *grey matter density* (GMD) and *white matter density* (WMD) (values closer to 1 indicate a high probability of a given tissue type in that voxel) and volumes (defined as the sum of the Jacobian determinants) were extracted for each of the ten functional parcellations (see Methods: Image pre-processing).

### Clinical validation of models using Social Responsiveness Scores (SRS)
We investigated whether deviations in cerebellar growth are present in children who are likely to fall on the Autism spectrum according to the *Social Responsiveness Scale* (SRS). A shortened 18-item version of the SRS was administered via questionnaire at the age of 8 years[68]. For each ROI separately, the z-scores of children likely to fall on the Autism spectrum (raw score on SRS >= 90th percentile, $n$ = 198) and the remainder of the cohort ($n$ = 2,012; children without SRS information excluded) were compared. We defined data as having large deviations in the normative model if the value of their normative estimate was larger than 1.96 or smaller than −1.96 (i.e., upper and lower tails of the distribution, critical z for 95% confidence interval). Due to the different sample sizes of typical and high SRS children, and therefore different expected proportions under the null hypothesis, significance of the percentage of children with large deviations at the $p$ = 0.05 level were evaluated using Binomial testing (observed vs. expected number of participants with $z > 1.96 / z < −1.96$ in high SRS and typical children, given a null hypothesized probability of $p_O$ = 0.025, one-sided). Crucially, given the heterogeneous brain morphology in Autism[69], the current approach does not rely on group-level inferences. By illustrating percentages of extreme deviations we can 1) validate that, as expected in a representative reference model, approximately 2.5% of typically developing children fall in the tails of the normative distribution, and 2) show whether children at risk for Autism diverge from this, even if individual patterns may show considerable variations. We additionally tested the effect of SRS score as a continues variable on the deviation scores using linear regression.

### Relating normative model deviations in the functional parcellation to IQ
Next to the clinical validation, we used the same approach to illustrate how IQ correlates with cerebellar deviations in the functional parcellation, which contains motor and cognitive subregions, in a supplementary analysis. Results were stratified based on IQ scores obtained from *Snijders-Oomen Nonverbal Intelligence Test* (SON-IQ) (please refer to *3.6. Non-response analysis* for details): Low IQ (<70; $n$ = 40), and high IQ (>130; $n$ = 64). We again defined data as having large deviations in the normative model if the value of their normative estimate was larger than 1.96 or smaller than −1.96 (i.e., upper and lower tails of the distribution, critical $z$ for 95% confidence interval) and significance of the percentage of children with large deviations at the $p$ = 0.05 and $p$ = 0.01 level was evaluated using Binomial testing (see Methods: *Clinical validation of models using Social Responsiveness Scores (SRS)* for details). We find global effects of lower volumes in children with low IQ along with large positive and negative deviations in GMD and WMD in posterior cognitive subregions (please see Supplementary Fig. 11 for details).

### Reporting summary
Further information on research design is available in the Nature Portfolio Reporting Summary linked to this article.

## Data availability
The cerebellar anatomical (https://github.com/CoBrALab/atlases)[7] and functional (https://github.com/DiedrichsenLab/cerebellar_atlases)[17] atlases are available on github. The cerebellar growth models have been deposited on github (https://github.com/cgaiser1/cerebellar-growth-models)[56] as well. The raw MRI and participant-level region-of-interest data are protected and are not available due to privacy laws. However, access can be requested via the Generation R administration (secretariaat.genr@erasmusmc.nl). Source data are provided with this paper.

## Code availability
Code to generate normative models and transfer knowledge from existing models to new sites is freely available via the PCNtoolkit (https://github.com/amarquand/PCNtoolkit).

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

## Acknowledgements

The authors thank Nadine Danner for support in quality control of the MRI images, Jonathan Krikeb for help with data management and Min Tae M. Park for assisting with implementation of the MAGeT anatomical segmentation algorithm. Importantly, we thank and are grateful for the contribution of children and parents of the Generation R study, as well as the researchers involved in data collection. The general design of Generation R Study is made possible by financial support from the Erasmus Medical Center, Rotterdam, the Erasmus University Rotterdam, ZonMw, The Netherlands Organization for Scientific Research (NWO), and the Ministry of Health, Welfare, and Sport. Image infrastructure and analysis were supported by the Sophia Foundation (S18-20), the Erasmus MC Fellowship, and the Dutch Scientific Organization (NWO, surfsara.nl, 2021.042) [R.M.]. The study was supported by the Erasmus MC² Research Synergy Grant [C.G., R.V., R.M., M.F.], the Health and Technology Convergence Alliance TU Delft, Erasmus[58] MC University Medical Center Rotterdam and Erasmus University Rotterdam [C.G., M.F.], the Wellcome Trust under an Innovator award ('BRAINCHART' 215698/Z/19/Z) [P.B., A.M.], the European Research Council, consolidator grant 'MENTALPRECISION' 10100118 [A.M.], and the Canadian Institutes of Health Research (CIHR, PJT 159520) [J.D.].

## Author contributions

C.G., R.V., M.F. and R.M.: originally conceived of the project. C.G. and R.M.: data curation and pre-processing. C.G., O.D., A.M. and R.M.: contributed to interpretation of results. A.B., P.B. and A.M.: provided expertise on normative modeling. G.D. and M.C.: provided expertise on the MAGeTBrain algorithm. J.D.: provided expertise on the functional parcellation of the cerebellum. M.F. and R.M.: supervision. C.G.: formal analysis, visualizations, and writing (original draft). C.G. and R.M.: wrote the manuscript with inputs from all authors.

## Competing interests

The authors declare no competing interests.
