## [Peer Review File · Nature Communications]

Population-wide Cerebellar Growth Models of Children and AdolescentsReviewer #1 (Remarks to the Author):

The authors propose an openly available model of cerebellar development in childhood and adolescence. The authors found an antero-posterior gradient related to age improvement. Finally they propose a clinical application of their model and propose to detect cerebellar abnormalities related to social abilities.

This work is very interesting and significant for the field. The cerebellum remains an understudied region of the brain. A large part of the cerebellum is involved in cognitive functions and might be involved in several psychiatric disorders with an onset during childhood or adolescence. Thus, understanding the growth of the cerebellum is important. This paper relies on the analysis of a large (n = 4 862) longitudinal cohort, acquired in only two MRI scanners. Previous papers on normative modeling did not study the cerebellum and this paper addresses a gap in the literature. The methods are detailed and well explained. I think that this paper would be very interesting to the scientific community and I recommend this paper for publication. I only have minor remarks.

1°) The authors conducted 2 different analyses with the Maget Brain atlas and with the MDTB atlas. The MNI-aligned version of the MDTB atlas was used. Could the authors explain why they use this version instead of the SUIT-aligned version of the MDTB atlas, given that SUIT space might be more suited for analyses in the cerebellum ?

2°) I think that it might be interesting for the reader to have just one visual example of a scan rated as "good", "sufficient" or "bad". This information could be included in the supplementary material.

3°) Line 565, "autistic traits" should be replaced by "autism"

4°) Line 525 : the authors discuss the absence of antero-posterior growth in the vermis. They suggest that this could be related to the role of the vermis in "lower-order" function. However several studies suggested that the vermis might be involved in emotion regulation and connected to limbic regions (see Fastenrath et al. 2022) :

<https://www.pnas.org/doi/10.1073/pnas.2204900119>

5°) The authors propose a clinical application to this work and detect cerebellar abnormalities related to social abilities (measured with the SRS scale). I think it might be interesting to provide a plot (histogram) describing the distribution of the SRS score in the population (which is, I believe, a non-clinical cohort). It would be interesting to know how many individuals had a high SRS-score in the cohort.

6°) Although this analysis could be optional, I think it might be interesting to provide additional data on the effect of IQ on cerebellar structure. For instance, I think that it might be interesting to study (i) if deviations from the normative models would be associated with either lower or higher IQ - most likely in the cognitive regions of the cerebellum and / or (ii) how the cerebellar growth during the development is associated with IQ score. I think the authors could report the min / max IQ scores in the cohort

Reviewer #2 (Remarks to the Author):

This study takes advantage of a large dataset (n=4862) of children and adolescents aged ~6-17 years to produce "growth curves" for the cerebellum. Both structural (segmentation into lobules) and functional (based on King et al. 2019) parcellations of the cerebellum were examined. This study fills a gap left by other studies that have investigated brain growth over the lifespan (e.g. Bethlehem et al. 2022) which did not evaluate cerebellar growth. Strengths include a focus on a sometimes-neglected neural structure that is implicated in various neurodevelopmental disorders and a wide range of functions, the use of both structural and functional parcellation approaches, and the availability of the models to other researchers. Limitations include a somewhat restricted age span (the earliest scans are from ~7 yrs old, after extensive brain growth has already occurred) and lack of clarity on aspects of the methods. The study uses an example of individuals with high vs. low scores on the Social Responsiveness Scale as a metric for evaluating the clinical utility of the models, but a dimensional approach may be a more rigorous test of how individual growth scores relate to behavioral metrics. A final limitation is that the authors did not take the opportunity to test the models with the longitudinal data to determine whether individuals stay on

their growth curve throughout the 3 timepoints measured during the study. This would demonstrate the utility of these models for tracking individual development, which is one argument for establishing such growth curves in the first place. Overall, there is certainly a need for better metrics regarding cerebellar development, particularly given cerebellar findings in a range of neurodevelopmental disorders and those with a history of preterm birth. Clarification of the methodological approach and testing of the models would strengthen the impact of this study.

- Title: (Very) minor suggestion: change "Large data" to "Big data" to match the more commonly used term for large datasets

- Abstract: Add in number of participants and the age range, since this study does not sample very early in development. This will be useful when researchers are searching for studies in particular age groups.

Methods:

- Aside from T1 image quality, were there any exclusion criteria for participants? E.g. preterm birth, neurodevelopmental diagnosis (ADHD, autism)?
- How were the 7 "unique and representative" images chosen from each time point?
- How was developmental age factored into the segmentation? Were the MAGeT images that were manually segmented from adults or pediatric populations?
- Note in the methods what the 35 parcellations refer to (e.g. hemispheric lobules III-VI, lobule VII (Crus I, Crus II, VIIB), VIIIA and B, etc.). It looks like there is a core WM parcellation as well, but that GM and WM are combined in the lobules?
- Were the functional parcellations performed in native space for each individual participant or in MNI space?
- It is not clear why the SRS scores were not evaluated (or also evaluated) as a continuous measure, or why there was not a "mid range" group as well as the high- and low-scoring group.
- In section 3.6, should "subsets" be "subtests"?
- The authors acknowledge the limitations of the gradient analysis. There are other gradient-based tools that could be used within the cerebellum, e.g. Guell et al. 2019 PlosOne "Little Brain".

Results

- The anterior lobe is lobules I-V, and VI is in the posterior lobe of the cerebellum
- Which of the regions show statistically significant changes with age in the age range examined?
- Fig 4 legend – I think this should read "Effect of age on volume in the functional parcellation"?
- Fig 4 – it is difficult to visualize the findings with the lobular boundaries in bold and the functional boundaries not demarcated. It would be clearer to either just demarcate the functional boundaries (given Fig 3 shows the lobular boundaries in the context of the functional boundaries) or lighten the lobular boundaries and demarcate the functional boundaries. As it is, it is difficult to see which functional regions show differential effects.
- Fig 4 – given the lobules are often labeled with roman numerals, perhaps use lowercase a, b, c etc for the individual panels.
- The authors aim to evaluate the clinical significance of the cerebellar growth curves by looking at the z-scores of individuals with the highest and lowest SRS scores. However, growth models are useful because they reveal the dimensionality of data and place individuals in this context. If SRS scores are used as a continuous variable, is there a relationship between z scores and SRS scores in any regions (structural or functional parcellation)?

Other comments

- The authors could acknowledge prior studies of cerebellar growth e.g. Tiemeier et al. 2010, Shaw et al. 2018 and others that have discussed functional gradients in the cerebellum e.g. Guell et al. 2018
- The authors acknowledge the limitations of the quantitative testing of the anterior-to-posterior gradient. An alternative would be to compare the growth coefficients between pairs of lobules (for anatomical parcellation) or motor vs. higher-order cognitive regions (e.g. right-hand presses and verbal fluency) in the functional parcellation.
- The data for waves 2 and 3 are heavily enriched in two age bands (wave 1 seems more evenly distributed between 7 and 10 yrs). How might this impact the accuracy of the growth curves for

ages where there is relatively little data available?

- It seems like a missed opportunity not to examine the subset of individuals with longitudinal data (scans at all 3 timepoints). Do these individuals stay on their growth curves over time?

Reviewer #3 (Remarks to the Author):

Here the authors examine normative developmental trajectories of the cerebellum and its constituent regions, something that has been lacking within human neuroscience for many years. The authors use a large neuroimaging dataset collected in the Netherlands whose longitudinal nature allows for powerful developmental analyses. They show that subsections of the cerebellum, whether defined anatomically or functionally, show distinct maturational rates, with interesting sex differences as well. Having created normative growth charts for regions of the cerebellum, the authors then go on to show that individuals with autism-related behavioral traits show significant deviation in several regions of the cerebellum, generally showing less cerebellar tissue. The authors provide a nice code-based framework for future researchers to take advantage of this large dataset. Overall these results are novel, statistically rigorous, and will be of interest to the field.

Figure 2, it would be helpful here to maybe have a bar graph or something similar showing the actual growth or %-change for each lobule. I like the renderings of the cerebellum, but having more quantitative representation alongside it so the reader can see what each lobule is doing would be useful. I don't think a bar graph would take up too much space and could be fit alongside the volume renderings. I ask because in Fig 2B, it seems as if Crus II and CM are increasing by almost 25% in volume from 7 to 17 years old, that seems like a lot! I think it's generally interpreted in the field that the brain (probably the cortex) has reached about 80% of its volume by the age of 5, so to show the cerebellum (or parts of it) is exceeding this would be something worth highlighting.

Individual images were linearly then nonlinearly aligned to a shared space. It later says that parcellations were done in native space. Can the author's clarify? After reading the methods section "Anatomical Parcellation" a few times, I think I finally understood what was happening. Five hand-segmented images are nonlinearly aligned to 21 representative brains spanning the age-range to create a small library of 105 segmented cerebella. Then an individual's brain is aligned to each of these 105 segmentations, and then a compression step happens (through voxel-wise majority voting) to get a final segmentation in an individual's brain. Is that correct? If not then this section might benefit from more explanation.

It would help to have a more intuitive description of what gray/white matter density are. Is it essentially how gray or white a given voxel is? Is it derived from the inherent voxel intensity within the individual's image, or is it derived by aligning the individual to the MNI space and is it extracted from some probability map there? This can be useful for interpretation. For example in Fig 4B it seems that GMD decreases and WMD increases, which might be in line with the interpretation that myelin content is increasing (thus whitening most voxels).

Figure 5 is nice. However, when ranking functional parcels onto the AP axis, the authors write that the volumetric/spatial centroid was used to determine in which anatomical lobule a given functional parcel falls. As the authors note, most functional parcels have two components (left-hand presses show representation in anterior and posterior lobules) and if the centroid falls outside of a lobule they hand-assign it to the nearest lobule. For the case of left-hand presses, that might be a lobule that doesn't actually have any left-hand press representation. Does that happen? The authors acknowledge that this approach isn't perfect. Maybe they could also just try a binary test comparing motor versus non-motor representations to test for gradient difference in development? Related to this point, Guell and colleagues in their "functional gradients of the cerebellum" article find an A-P gradient (gradient 1) from functional connectivity, maybe using those values to assign your functional parcels a better sensory-cognitive score (rather than a spatial centroid) would

help? The sex difference flatmaps in Figure 4A for GMD and WMD seem to map onto the Guell gradient 1 map quite well (peaks near Crus1/2, low values in anterior lobules like LobI-III and IX) so it could be a worthwhile endeavor. Also, and perhaps this is just a semantic difference, but isn't lobule IX as anterior as lobule I-III? Isn't Crus I/II the most posterior?

Within the field of neuroimaging, the volumetric findings from autism cohorts on the brain and cerebellum are a little all over the place, so in some sense the authors' findings using this massive dataset (the largest study of the cerebellum to date) stand to add a lot of clarity. I think the authors could increase their discussion on this point, pointing out which studies they replicate, and perhaps discussing that their parcellated approach (rather than treating the cerebellum as a single structure) perhaps allowed for better sensitivity.

Minor Comments:

CCM is used as the unit for volume analyses (e.g., Figure 2B). What does it stand for? Is it cubic centimeters? I'm not sure the acronym is defined anywhere.

Seems like Fig 3 could be added as an inset into Fig 4 given how related they are (and that 3 is just a reproduction).

Reviewer #4 (Remarks to the Author):

1. What are the noteworthy results?

Comment: The first normative model of anatomical and functional subregions of the cerebellum was established from a large pediatric population, and revealed an anterior-posterior gradient of human cerebellum.

2. Will the work be of significance to the field and related fields? How does it compare to the established literature? If the work is not original, please provide relevant references.

Comment: The present work is highly novel and holds great significance to the field of human brain mapping as well as related fields such as brain disorders and public health. Some novel results (e.g., the A-P gradient of brain growth is highly implicated with previous literature on brain maturation such as cortical gradient development).

3. Does the work support the conclusions and claims, or is additional evidence needed?

Comment: Overall, the work has been done in a very solid way. But, it would be more informative if the authors can demonstrate some valid usage of the proposed normative models, for example, the association studies on the cerebellum and behavior/cognition.

4. Are there any flaws in the data analysis, interpretation and conclusions? Do these prohibit publication or require revision?

Is the methodology sound? Does the work meet the expected standards in your field?

Comment: A problematic point is about the unequal distribution of sample ages, which could be the driving force of the overall linear trajectories modeled. The authors need to be careful for the interpretation on their findings while they can do some validation based on some smaller-scale cohorts such as Healthy Brain Network or Chinese Color Nest. These cohorts are openly shared to the community.

5. Is there enough detail provided in the methods for the work to be reproduced?

Comment: Yes.

6. Other Comments: The scientific finding of the A-P growth gradient is very interesting while can be further informed with recent advances on developmental shifts of cortical connectivity gradient

from childhood to adolescence, please refer to <https://pubmed.ncbi.nlm.nih.gov/34260385>.

Response to the Reviewers

We are grateful to the editor and reviewers for their constructive and helpful comments. We feel the changes and additions as a result of these suggestions have helped to improve the manuscript considerably.

Reviewer #1 (Remarks to the Author):

The authors propose an openly available model of cerebellar development in childhood and adolescence. The authors found an antero-posterior gradient related to age improvement. Finally they propose a clinical application of their model and propose to detect cerebellar abnormalities related to social abilities.

This work is very interesting and significant for the field. The cerebellum remains an understudied region of the brain. A large part of the cerebellum is involved in cognitive functions and might be involved in several psychiatric disorders with an onset during childhood or adolescence. Thus, understanding the growth of the cerebellum is important. This paper relies on the analysis of a large ($n = 4\,862$) longitudinal cohort, acquired in only two MRI scanners. Previous papers on normative modeling did not study the cerebellum and this paper addresses a gap in the literature.

The methods are detailed and well explained. I think that this paper would be very interesting to the scientific community and I recommend this paper for publication. I only have minor remarks.

1) The authors conducted 2 different analyses with the Maget Brain atlas and with the MDTB atlas. The MNI-aligned version of the MDTB atlas was used. Could the authors explain why they use this version instead of the SUIT-aligned version of the MDTB atlas, given that SUIT space might be more suited for analyses in the cerebellum ?

It is indeed correct, that normalization to the SUIT atlas compared to the former linear MNI template has been shown to improve overlap of cerebellar regions across individuals (Diedrichsen, 2006). However, there are several reasons why we decided that using the MNI-aligned version of the MDTB atlas outweighs the added benefit of better alignment of some of the cerebellar anatomy (mainly cerebellar fissures).

First and foremost, the primary aim of this study is to provide openly available, large-scale cerebellar growth models that, we hope, increase and facilitate cerebellar neuroimaging research, also outside of the cerebellar research community. By using the MNI template, future research groups can potentially utilize the cerebellar models without having to apply additional image processing steps. This is because commonly used pre-processing tools (i.e. FMRIPrep, FSL, SPM) already include normalization to standard MNI space, which clears the way to incorporate the MNI-aligned version of the MDTB atlas in a straightforward way. Second, we used SMRIPREP (and ANTs) to conduct the nonlinear registration given it is a.) open source (i.e., no Matlab license required) and b.) the ANTs registration methods are used widely in the field given their

superior performance with structural image co-registration. Importantly, the better alignment with the linear MNI space had to do with the non-linear morphs used in the atlas generation of SUIT. However, alignment to the MNI152NLin (the non-linear template MNI template published since) has greatly improved MNI normalization. Further, MNI space is very similar to SUIT space and coordinates can be used in both spaces interchangeably with only very minor morphs (Diedrichsen, 2006).

We have addressed this in the manuscript:

Methods, Functional parcellation (lines 703-708)

The MNI-aligned atlas was chosen over the SUIT space aligned atlas, since it allows to use the current models without adding an additional processing step to transform images into SUIT space. Furthermore, while normalization to SUIT space compared to the former linear MNI template has been shown to improve overlap of cerebellar regions across individuals (Diedrichsen, 2006), alignment to the MNI152NLin (the non-linear template MNI template published since) has greatly improved MNI normalization since then.

2) I think that it might be interesting for the reader to have just one visual example of a scan rated as “good”, “sufficient” or “bad”. This information could be included in the supplementary material.

Thank you for the suggestion. We included a figure in the supplementary material showing examples of scans rated as either “Good”, “Sufficient”, or “Bad”.

We added in the manuscript:

Methods, Image Quality Control (lines 671-672)

Visual examples of scans rated on the 3-point scale can be found in Supplementary Figure 6.

Supplementary Figures (lines 980-983)

Supplementary Figure 6: Visual examples of segmented scans rated as either “good”, “sufficient”, or “bad”. Inaccuracies of only a few voxels were rated as “sufficient” (e.g., vasculature was captured), whereas marked inaccuracies throughout several parcels were rated as “bad” and therefore excluded from analysis.

3) Line 565, “autistic traits” should be replaced by “autism”

Thank you, we changed the manuscript accordingly.

We changed in the manuscript:

Discussion (lines 471-473)

Recently, however, no differences in cerebellar anatomy in individuals with autism were reported when using normative models on cerebellar growth based on a smaller control sample (N=219) (Laidi et al., 2022).

4) Line 525 : the authors discuss the absence of antero-posterior growth in the vermis. They suggest that this could be related to the role of the vermis in “lower-order” function. However several studies suggested that the vermis might be involved in emotion regulation and connected to limbic regions (see Fastenrath et al. 2022) : <https://www.pnas.org/doi/10.1073/pnas.2204900119>

While the vermis has been found to be predominantly involved in somatosensory, ocular, and vestibular functions with strong connections with the spinal cord, it is indeed correct that functionally, the vermis is not exclusively involved in lower-order functions only. Thank you for raising that point, we now specify this in the manuscript.

We added in the manuscript:

Discussion (lines 413-415)

The vermis receives sensorimotor afferents coming from the spinal cord, and is predominantly involved in lower-order functions, such as postural control, locomotion, and gaze (Kandel et al., 2000), but also plays an important role in emotion processing (Baumann & Mattingley, 2012; Fastenrath et al., 2022; Sacchetti et al., 2009).

5) The authors propose a clinical application to this work and detect cerebellar abnormalities related to social abilities (measured with the SRS scale). I think it might be interesting to provide a plot (histogram) describing the distribution of the SRS score in the population (which is, I believe, a non-clinical cohort). It would be interesting to know how many individuals had a high SRS-score in the cohort.

Thank you for the suggestion. The Generation R cohort is indeed a non-clinical cohort, therefore most of the participants do not have high SRS scores. We now also include a histogram of the distribution of all available SRS scores in the cohort to give the reader a better insight into the data, and we also include the number of children with a high SRS score (defined as being above the 90th percentile) in the methods section.

We added in the manuscript:

Results, Large normative model deviations and clinical or behavioral phenotypes (lines 331-334)

For each ROI, we contrast the z-scores between children likely to fall on the Autism spectrum (raw score on SRS \geq 90th percentile, $N = 198$) to the remainder of the cohort ($N = 2,012$; children without SRS information excluded). The distribution of SRS scores can be found in Supplementary Figure 4.

Supplementary Figures (lines 822-835)

Supplementary Figure 4: Distribution of SRS scores in the Generation R cohort ($N = 2,210$; participants without SRS information excluded). The dotted line illustrates the 90th percentile (raw score ≥ 9 [$N = 198$]), triangles draw attention to high SRS scores of single participants.

6) Although this analysis could be optional, I think it might be interesting to provide additional data on the effect of IQ on cerebellar structure. For instance, I think that it might be interesting to study (i) if deviations from the normative models would be associated with either lower or higher IQ - most likely in the cognitive regions of the cerebellum and / or (ii) how the cerebellar growth during the development is associated with IQ score. I think the authors could report the min / max IQ scores in the cohort

The association between IQ scores and cerebellar deviations is definitely an interesting analysis, specifically in the functional parcellation where several parcels represent cognitive influences. We therefore added an analysis contrasting cerebellar deviations in the functional parcellation between children with high (above 130) and low (below 70) IQ. We found that children with lower IQ have lower cerebellar volumes, specifically in anterior and posterior parcels of the right cerebellar hemisphere, as well as more deviations in GMD and WMD in posterior attention and language parcels. We thank the reviewer for this valuable suggestion and have added the analysis in the supplement.

We added in the manuscript:

Methods, Relating normative model deviations in the functional parcellation to IQ (lines 731-743)

Next to the clinical validation, we used the same approach to illustrate how IQ correlates with cerebellar deviations in the functional parcellation, which contains motor and cognitive subregions, in a supplementary analysis. Results were stratified based on IQ scores obtained from Snijders-Oomen Nonverbal Intelligence Test (SON-IQ) (please refer to 3.6. Non-response analysis for details): Low IQ (< 70; N=40), and high IQ (>130; N=64). We again defined data as having large deviations in the normative model if the value of their normative estimate was larger 1.96 or smaller -1.96 (i.e., upper and lower tails of the distribution, critical z for 95% confidence interval) and significance of the percentage of children with large deviations at the $p = 0.05$ and $p = 0.01$ level was evaluated using Binomial testing (see Methods: Clinical validation of models using Social Responsiveness Scores (SRS) for details). We find global effects of lower volumes in children with low IQ along with large positive and negative deviations in GMD and WMD in posterior cognitive subregions (please see Supplementary Figure 8 for details).

Supplementary Figures (lines 991-1006)

Supplementary Figure 8: Percentage of individuals with large negative (z-score < -1.96) and large positive (z-score > 1.96) deviations in volume, *Grey Matter Density* (GMD), and *White Matter Density* (WMD) in functional ROIs stratified by IQ (low IQ <70 [N=40] and high IQ >130 [N=64]). As sample sizes differ between IQ groups, and thus expected proportions of extreme deviations under the null hypothesis differ as well, significance of the percentage of children with large deviations at the $p = 0.05$ and $p = 0.01$ level were evaluated using Binomial testing (observed vs. expected number of participants with $z > 1.96 / z < -1.96$ in low, typical, and high IQ children, given a null hypothesized probability of $p_0 = 0.025$, one-sided). Asterisks ($p < 0.05$) and stars ($p < 0.01$) indicate ROIs in which children have a significantly higher percentages of large deviation than expected (low IQ > 5.00% ($p < 0.05$) and > 7.50% ($p < 0.01$), high IQ > 3.12% ($p < 0.05$) and > 6.25% ($p < 0.01$)). Children in the low IQ group present with lower volumes than expected throughout several ROIs particularly on the right hemisphere (negative deviations (at $p < 0.01$ level): 2 *Right-hand (motor) presses*, 3 *Saccades*, 4 *Action observation*, 6 *Divided attention (right hemisphere)*, 8 *Word comprehension*, and 9 *Verbal fluency*). Lower IQ was further associated with more negative as well as positive deviations in GMD (negative deviations (at $p < 0.01$ level): 5 *Divided attention (left hemisphere)*); positive deviations (at $p < 0.01$ level): 8 *Word comprehension*, and 9 *Verbal fluency*) and WMD (negative deviations (at $p < 0.01$ level): 8 *Word comprehension*, and 9 *Verbal fluency*; positive deviations (at $p < 0.01$ level): 1 *Left-hand (motor) presses*, 5 *Divided attention (left hemisphere)*, 6 *Divided attention (right hemisphere)*) specifically in posterior ROIs relating to cognitive function.

Reviewer #2 (Remarks to the Author):

This study takes advantage of a large dataset (n=4862) of children and adolescents aged ~6-17 years to produce “growth curves” for the cerebellum. Both structural (segmentation into lobules) and functional (based on King et al. 2019) parcellations of the cerebellum were examined. This study fills a gap left by other studies that have investigated brain growth over the lifespan (e.g. Bethlehem et al. 2022) which did not evaluate cerebellar growth. Strengths include a focus on a sometimes-neglected neural structure that is implicated in various neurodevelopmental disorders and a wide range of functions, the use of both structural and functional parcellation approaches, and the availability of the models to other researchers. Limitations include a somewhat restricted age span (the earliest scans are from ~7 yrs old, after extensive brain growth has already occurred) and lack of clarity on aspects of the methods. The study uses an example of individuals with high vs. low scores on the Social Responsiveness Scale as a metric for evaluating the clinical utility of the models, but a dimensional approach may be a more rigorous test of how individual growth scores relate to behavioral metrics. A final limitation is that the authors did not take the opportunity to test the models with the longitudinal data to determine whether individuals stay on their growth curve throughout the 3 timepoints measured during the study. This would demonstrate the utility of these models for tracking individual development, which is one argument for establishing such growth curves in the first place. Overall, there is certainly a need for better metrics regarding cerebellar development, particularly given cerebellar findings in a range of neurodevelopmental disorders and those with a history of preterm birth. Clarification of the methodological approach and testing of the models would strengthen the impact of this study.

1) Title: (Very) minor suggestion: change “Large data” to “Big data” to match the more commonly used term for large datasets

Thank you for the comment, we have changed the title accordingly.

We changed in the manuscript:

Title (line 1)

Big Data on the Small Brain: Population-wide Cerebellar Growth Models of Children and Adolescents

2) Abstract: Add in number of participants and the age range, since this study does not sample very early in development. This will be useful when researchers are searching for studies in particular age groups.

Thank you for the suggestion, we added the number of participants and age range in the abstract.

We added in the manuscript:

Abstract (lines 46-48)

Using a total of 7,240 neuroimaging scans from 4,862 individuals, we describe and provide detailed, openly available models of cerebellar development in childhood and adolescence (age range: 6-17 years), an important time period for brain development and onset of neuropsychiatric disorders.

Methods:

3) Aside from T1 image quality, were there any exclusion criteria for participants? E.g. preterm birth, neurodevelopmental diagnosis (ADHD, autism)?

No, we only applied the exclusion criteria stated in the manuscript. Scans were excluded if 1) the T₁-weighted scan was incomplete, 2) no consent form was present, 3) incidental finding was present, and/or 4) quality rating was insufficient. We chose to not exclude children born preterm and/or with neurodevelopmental disorders in order to have models that are representative and cover the heterogeneity of the general population. Therefore, we restricted our exclusion criteria to only remove scans, if they are likely to result in segmentation errors (incomplete/low quality, incidental findings like increased ventricles or large cysts) or due to ethical reasons (incomplete consent forms).

We added in the manuscript:

Methods, Participants (lines 593-595)

Scans were only excluded based on technical or ethical considerations, but not based on clinical phenotypes (i.e. pre-existing conditions) in order to capture the heterogeneity of the general population.

4) How were the 7 “unique and representative” images chosen from each time point?

Thank you for the comment. This is indeed useful for the reader to know, so we have adapted the text as follows:

We added in the manuscript:

Methods, Anatomical parcellation (lines 682-687)

Seven unique and representative images from the three study time points were selected, by a) excluding scans with dental implants, b) pre-selecting the top 20 scans with the highest quality ratings based on automatic FreeSurfer quality assurance (Rosen et al., 2018), and c) thoroughly inspecting the top 20 scans per time point for artifacts, inhomogeneities, full coverage of the cerebellum, and (cerebellar) cysts. Based on this evaluation, the 7 scans with the highest quality ratings were selected per time point, resulting in 21 study-specific template images.

5) How was developmental age factored into the segmentation? Were the MAGeT images that were manually segmented from adults or pediatric populations?

The five manually segmented MAGeT atlases stem from a healthy adult population without neurological/neuropsychiatric disorders (2 male, 3 female, aged 29–57) (Park et al., 2014). The strength of the MAGeT algorithm lies in its ability to create a multi-atlas based segmentation, tailored to your specific cohort (template images) using only a limited amount of manually segmented atlases (here: five). The resulting template library, allows for improved modeling of individual differences in morphology (in native space), which could be present in a pediatric compared to an adult population, by taking advantage of the morphological variations in the representative template images. This approach has been used successfully in a pediatric cohort before (Shaw et al., 2018) and visual inspection showed high accuracy of lobular segmentations in our cohort (94.8% useable). We now emphasize this strength in the methods section but also added the use of adult atlases as a potential limitation for consideration, since this is an important and recurring theme in (developmental) neuroimaging studies.

We added in the manuscript:

Methods, Anatomical parcellation (line 697-700)

This computationally intensive approach has been shown to have better test-retest reliability than other segmentation techniques and results in high segmentation accuracy by creating a customizable template segmentation library, thereby being able to take advantage of existing morphological variances in the cohort (Park et al., 2014).

Discussion, Limitations (lines 548-552)

Another potential limitation is related to how the MRI data were processed, namely the use of an adult template space and adult atlases. This is a recurring theme in human neuroimaging, which has yet to be fully resolved. Importantly, the MAGeTBrain framework is likely able to improve modelling of individual differences in morphology, possibly present in a developing cohort compared to an adult population, through the use of study-specific template propagation.

6) Note in the methods what the 35 parcellations refer to (e.g. hemispheric lobules III-VI, lobule VII (Crus I, Crus II, VIIB), VIIIA and B, etc.). It looks like there is a core WM parcellation as well, but that GM and WM are combined in the lobules?

Thank you for the comment, we now provide a better overview of the anatomical subdivisions in the manuscript. The corpus medullare is segmented as the core white matter for each hemisphere. The white matter branching out into the lobules was included in the segmentation of the lobules, since ultra-high resolution scans are needed to reliably segment the fine white matter branches reaching out into the cerebellar folia (Marques et al., 2010).

We added in the manuscript:

Methods, Anatomical parcellation (line 696-697)

Supplementary Figure 7 shows a representative automatically labelled segmentation from one individual and gives an overview of all 35 anatomical subdivisions.

Supplementary Figures (lines 985-989)

Supplementary Figure 7: Example scan that was automatically segmented in anatomical ROIs using the MAGeT algorithm. Labels for each ROI in their respective colors are shown. The MAGeT algorithm subdivides the cerebellum into 11 vermal and 22 hemispheric lobules (11 on each hemisphere). Additionally, the central white matter, the corpus medullare, is segmented in each hemisphere. White matter that extends into the folia of the lobules was segmented as part of the lobules.

7) Were the functional parcellations performed in native space for each individual participant or in MNI space?

Functional parcels of the MDTB atlas were performed in MNI space. We now emphasize this in the manuscript.

We added in the manuscript:

Methods, Functional parcellation (lines 702&703)

We parcellated functional subregions of the cerebellum in MNI space, using the MNI-aligned version of the MDTB atlas (King et al., 2019).

8) It is not clear why the SRS scores were not evaluated (or also evaluated) as a continuous measure, or why there was not a “mid range” group as well as the high- and low-scoring group.

The normative modeling concept has been proposed as a route for individual-level inferences to be made, and thus we focused on highlighting this in the manuscript by including example applications of such (i.e., percentage of children with extreme deviations). Given the non-normal

distribution of the SRS scores, we decided not to create a mid-range group. However, we agree that the dimensional nature of ASD traits and brain structure is generally of interest and have added an analysis of continuous SRS scores. We refer to question 16 below on the incorporation of continuous scores.

9) In section 3.6, should “subsets” be “subtests”?

This should indeed be subtests. Thank you for bringing it to our attention, we have corrected this typo.

10) The authors acknowledge the limitations of the gradient analysis. There are other gradient-based tools that could be used within the cerebellum, e.g. Guell et al. 2019 PlosOne “Little Brain”.

We thank the reviewer for this suggestion and valuable addition to the manuscript. We added an comparison between the gradient found with the gradients described by Guell and colleagues (Guell et al., 2019). We refer to comment 18 for details.

Results

11) The anterior lobe is lobules I-V, and VI is in the posterior lobe of the cerebellum

Apologies, for this typographical error. Thank you for making us aware of this mistake.

We changed the manuscript to remedy this:

Results, Anatomical parcellation (lines 194-197)

Interestingly, we see a growth gradient, starting with smaller age-related effects on volume in the anterior cerebellum (Lobules III – V), and increasingly larger age-related effects in the posterior cerebellum (Lobules VI – IX) with the largest effects, besides the corpus medullare, found in the flocculus (Lobules X).

12) Which of the regions show statistically significant changes with age in the age range examined?

The mean standardized β age, or slopes, of all ROIs are statistically significant as the confidence interval does not include 0 for any of the ROIs (please refer to Supplementary Tables 3&4). However, given the large sample size and high precision in measurements, it is also crucial to take the effect sizes into account (Marek et al., 2022). Standardized coefficients provide a measure for investigating the magnitude of the effect across different atlases (anatomical vs. functional), modalities (volume vs. GMD vs. WMD), and can also be used to compare effects across different studies. We therefore advice to judge the age related changes based on the standardized age coefficient.

We added in the manuscript:

Supplementary Table 3, Legend

While all slopes are significantly different from 0, some effects (standardized coefficients) are small.

Supplementary Table 4, Legend

While all slopes are significantly different from 0, some effects (standardized coefficients) are very small (e.g. GMD hand presses).

13) Fig 4 legend – I think this should read “Effect of age on volume in the functional parcellation”?

Thank you for pointing this out, this is indeed an oversight on our part. Please see response to comment 1 of reviewer 3 for the updated figure legend of figure 3 (previously figure 4).

14) Fig 4 – it is difficult to visualize the findings with the lobular boundaries in bold and the functional boundaries not demarcated. It would be clearer to either just demarcate the functional boundaries (given Fig 3 shows the lobular boundaries in the context of the functional boundaries) or lighten the lobular boundaries and demarcate the functional boundaries. As it is, it is difficult to see which functional regions show differential effects.

We thank the reviewer for this suggestion. We have implemented this by removing the lobular boundaries from the figures showing the functional cerebellar flatmap, in order not to mask the functional regions. See now main figures 3 and 5 as well as supplementary figures 5 and 8 in the manuscript, or also in our response to the a) 6th comment of reviewer 1 and b) first comment of reviewer 3.

15) Fig 4 – given the lobules are often labeled with roman numerals, perhaps use lowercase a, b, c etc for the individual panels.

We thank the reviewer for this suggestion, and we changed the panel numbers to lowercase letters. The updated figure can be found in the manuscript, and also in our response to comment 1 of reviewer 3.

16) The authors aim to evaluate the clinical significance of the cerebellar growth curves by looking at the z-scores of individuals with the highest and lowest SRS scores. However, growth models are useful because they reveal the dimensionality of data and place individuals in this context. If SRS scores are used as a continuous variable, is there a relationship between z scores and SRS scores in any regions (structural or functional parcellation)?

Thank you for this comment. As the reviewer correctly points out, the great strength of the normative growth models is that conclusions on morphological differences can be drawn on a single-subject basis - an approach particularly useful in disorders that present with great heterogeneity in the population, like, for example, autism. However, the reviewer also points out the interest in the dimensional nature of autistic traits, and whether or not they covary with neuroanatomical phenotypes, which has been shown previously. We have thus added new analysis where a linear regression was used to probe such associations. Interestingly, we find that none of the initial associations between deviation scores in posterior lobules and SRS scores survive multiple testing correction (multiple testing correction was applied separately for anatomical ROIs (N=35) and functional ROIs (N=30)). Yet, when using the current approach of investigating z-scores in children likely to fall on the autism spectrum, deviation patterns that are corroborated by previous findings (see Discussion) emerge. We added the linear regression results in the revised manuscript.

We added in manuscript:

Methods, Clinical validation of models using Social Responsiveness Scores (SRS) (lines 729&730)

We additionally tested the effect of SRS score as a continuous variable on the deviation scores using linear regression.

Results, Large normative model deviations and clinical or behavioral phenotypes (lines 369-374)

When the effect of SRS scores on normative deviation scores in both the anatomical and the functional parcellation was tested using linear regression models, volumes in anatomical lobules Crus II (hemispheric right) , VIIB (hemispheric left and right), VIIIA (hemispheric left) , and VIIIB (vermal) as well as functional parcels *action observation*, *word comprehension* and *autobiographical recall* were associated with SRS scores. However, these associations did not survive multiple testing correction (**Supplementary Table 6**).

ROI	SRS score				Square-root transformed SRS score				R ² adjusted
	Estimate	SE	p-value	FDR corrected p-value	Estimate	SE	p-value	FDR corrected p-value	
Vermis I & II	-0.223	0.220	0.309	0.994	0.055	0.217	0.799	0.874	0.001
Lobule III (left)	-0.199	0.214	0.353	0.994	-0.007	0.211	0.973	0.973	0.002
Lobule IV (left)	-0.261	0.214	0.223	0.994	0.040	0.211	0.851	0.903	0.002
Lobule V (left)	-0.005	0.217	0.982	0.994	-0.242	0.214	0.258	0.437	0.003
Lobule VI (left)	-0.115	0.220	0.600	0.994	-0.217	0.217	0.317	0.462	0.005
Lobule Crus 1 (left)	-0.089	0.218	0.682	0.994	-0.283	0.215	0.188	0.391	0.007
Lobule Crus 2 (left)	0.128	0.214	0.551	0.994	-0.412	0.211	0.052	0.208	0.004
Lobule VIIIB (left)	0.229	0.213	0.281	0.994	-0.650	0.210	0.002*	0.068	0.011
Lobule VIIIA (left)	0.141	0.214	0.511	0.994	-0.602	0.211	0.004*	0.077	0.013
Lobule VIIIB (left)	0.006	0.222	0.979	0.994	-0.311	0.219	0.156	0.389	0.004
Lobule IX (left)	-0.187	0.212	0.379	0.994	-0.106	0.210	0.613	0.716	0.004
Lobule X (left)	0.087	0.220	0.692	0.994	-0.371	0.217	0.087	0.277	0.004
Corpus medullare (left)	-0.050	0.216	0.816	0.994	-0.184	0.213	0.388	0.543	0.002
Vermis III	-0.188	0.214	0.379	0.994	0.067	0.211	0.752	0.849	0.000
Vermis IV	0.202	0.222	0.363	0.994	-0.421	0.219	0.054	0.208	0.003
Vermis V	-0.180	0.217	0.408	0.994	0.127	0.214	0.552	0.666	-0.001
Vermis VI	-0.002	0.223	0.994	0.994	-0.221	0.220	0.316	0.462	0.002
Vermis VIIA	0.330	0.224	0.141	0.994	-0.440	0.221	0.047	0.208	0.001
Vermis VIIB	0.212	0.221	0.337	0.994	-0.339	0.218	0.121	0.352	0.001
Vermis VIIIA	-0.080	0.215	0.712	0.994	-0.317	0.212	0.136	0.365	0.008
Vermis VIIIB	0.044	0.214	0.837	0.994	-0.459	0.211	0.030*	0.208	0.009
Vermis IX	-0.051	0.215	0.813	0.994	-0.270	0.212	0.202	0.393	0.005
Vermis X	-0.147	0.218	0.502	0.994	-0.011	0.215	0.959	0.973	0.000
Lobule III (right)	-0.013	0.214	0.953	0.994	-0.216	0.211	0.308	0.462	0.002
Lobule IV (right)	-0.068	0.221	0.759	0.994	-0.174	0.218	0.425	0.572	0.002
Lobule V (right)	-0.147	0.220	0.504	0.994	-0.141	0.217	0.515	0.654	0.003
Lobule VI (right)	-0.075	0.218	0.732	0.994	-0.282	0.215	0.190	0.391	0.006
Lobule Crus 1 (right)	0.016	0.213	0.939	0.994	-0.281	0.210	0.182	0.391	0.003
Lobule Crus 2 (right)	0.248	0.213	0.244	0.994	-0.527	0.210	0.012*	0.142	0.005
Lobule VIIIB (right)	0.028	0.217	0.899	0.994	-0.454	0.214	0.034*	0.208	0.010
Lobule VIIIA (right)	-0.071	0.215	0.741	0.994	-0.400	0.212	0.059	0.208	0.012
Lobule VIIIB (right)	0.027	0.220	0.901	0.994	-0.418	0.217	0.055	0.208	0.008
Lobule IX (right)	-0.215	0.217	0.323	0.994	-0.137	0.214	0.524	0.654	0.006
Lobule X (right)	-0.080	0.221	0.717	0.994	-0.261	0.218	0.231	0.426	0.005
Corpus medullare (right)	-0.058	0.218	0.791	0.994	-0.241	0.215	0.262	0.437	0.004
left hand presses (VOL)	-0.145	0.217	0.503	0.994	-0.240	0.214	0.262	0.576	0.007
right hand presses (VOL)	-0.110	0.217	0.612	0.994	-0.237	0.214	0.269	0.576	0.006
saccades (VOL)	-0.123	0.222	0.580	0.994	-0.264	0.219	0.229	0.576	0.007
action observation (VOL)	0.041	0.214	0.849	0.994	-0.516	0.211	0.014*	0.164	0.013
divided attention (left) (VOL)	0.023	0.218	0.915	0.994	-0.382	0.215	0.076	0.327	0.006
divided attention (right) (VOL)	-0.002	0.216	0.994	0.994	-0.392	0.213	0.066	0.327	0.008
narrative (VOL)	-0.069	0.214	0.748	0.994	-0.196	0.212	0.355	0.592	0.003
word comprehension (VOL)	0.200	0.215	0.351	0.994	-0.509	0.212	0.016*	0.164	0.006
verbal fluency (VOL)	0.041	0.216	0.850	0.994	-0.375	0.213	0.078	0.327	0.006
autobiographical recall (VOL)	0.146	0.215	0.496	0.994	-0.577	0.212	0.007*	0.164	0.011
left hand presses (GMD)	-0.105	0.217	0.630	0.994	-0.088	0.214	0.682	0.854	0.001
right hand presses (GMD)	-0.150	0.215	0.483	0.994	0.052	0.212	0.804	0.909	0.000
saccades (GMD)	-0.066	0.222	0.768	0.994	0.009	0.219	0.966	0.966	-0.001
action observation (GMD)	-0.011	0.224	0.961	0.994	-0.187	0.220	0.396	0.626	0.001
divided attention (left) (GMD)	0.027	0.219	0.902	0.994	-0.223	0.216	0.302	0.582	0.001
divided attention (right) (GMD)	-0.122	0.217	0.572	0.994	-0.016	0.214	0.942	0.966	0.000
narrative (GMD)	0.041	0.222	0.853	0.994	-0.075	0.219	0.734	0.880	-0.001
word comprehension (GMD)	0.234	0.223	0.294	0.994	-0.254	0.220	0.248	0.576	0.000
verbal fluency (GMD)	-0.024	0.221	0.913	0.994	-0.089	0.218	0.683	0.854	0.000
autobiographical recall (GMD)	0.078	0.229	0.734	0.994	-0.217	0.225	0.335	0.591	0.000
left hand presses (WMD)	0.052	0.220	0.814	0.994	0.147	0.217	0.499	0.712	0.001
right hand presses (WMD)	0.052	0.218	0.812	0.994	0.037	0.215	0.864	0.926	0.000
saccades (WMD)	-0.004	0.220	0.986	0.994	0.050	0.217	0.819	0.909	-0.001
action observation (WMD)	-0.149	0.221	0.501	0.994	0.368	0.218	0.092	0.327	0.002
divided attention (left) (WMD)	-0.152	0.220	0.490	0.994	0.359	0.217	0.098	0.327	0.002
divided attention (right) (WMD)	0.021	0.220	0.924	0.994	0.156	0.217	0.474	0.711	0.001
narrative (WMD)	-0.081	0.222	0.716	0.994	0.125	0.219	0.570	0.777	-0.001
word comprehension (WMD)	-0.241	0.224	0.281	0.994	0.282	0.221	0.201	0.576	0.000
verbal fluency (WMD)	-0.069	0.225	0.761	0.994	0.225	0.222	0.310	0.582	0.001
autobiographical recall (WMD)	-0.297	0.225	0.186	0.994	0.419	0.222	0.059	0.327	0.001

Supplementary Table 6: Linear regression results investigating the effect of SRS score (continuous) on deviation scores. SRS scores and square-root transformed SRS scores (given the skewed distribution, see **Supplementary Figure 4**) were used as the

independent variables. To account for the multiple tests, we applied the false discovery rate – Benjamini Hochberg (FDR-BH) correction within each parcellation separately.

Other comments

17) The authors could acknowledge prior studies of cerebellar growth e.g. Tiemeier et al. 2010, Shaw et al. 2018 and others that have discussed functional gradients in the cerebellum e.g. Guell et al. 2018

Thank you for the comment. We have included a discussion on previous studies of typical cerebellar development of Tiemeier (2010) and Shaw (2018). We further added additional information and an extra analysis covering the functional gradients proposed by Guell (2018). For additions concerning Guell (2018), please refer to comment 18.

We added in manuscript:

Discussion (lines 503-507)

Previous investigations into the typical volumetric development of the cerebellum during childhood and adolescence, although conducted with smaller sample sizes and employing less detailed anatomical parcellations, have revealed similar growth patterns. Increases were most pronounced in the corpus medullare and superior posterior lobe, while the anterior lobe and vermal regions showed stagnation or decline (Shaw et al., 2018; Tiemeier et al., 2010)

18) The authors acknowledge the limitations of the quantitative testing of the anterior-to-posterior gradient. An alternative would be to compare the growth coefficients between pairs of lobules (for anatomical parcellation) or motor vs. higher-order cognitive regions (e.g. right-hand presses and verbal fluency) in the functional parcellation.

We think this is an excellent addition to the manuscript. We have added an additional analysis using the LittleBrain toolbox (<https://xaviergp.github.io/littlebrain/>) to investigate the anterior-posterior growth gradient based on the reviewer's previous suggestion (comment #10). Using this toolbox, we illustrate the anterior-posterior growth along two principal functional gradients (Guell et al., 2018). The first functional gradient is essentially mapping cerebellar voxels along a continuum from motor to non-motor (mainly default mode and language) functions. While the reported anterior-posterior growth follows gradient 1 to some extent as well, it maps more closely onto gradient 2 which ranges from task unfocused areas with low cognitive load (motor, default parcels) to task focused areas with higher cognitive load (working memory) (see Figure 4B). We now consider the results of this analysis in the discussion as well.

We have adapted the manuscript as follows:

Results, Anterior-posterior growth gradient (lines 296-325)

Lastly, we compared and visualized the gradients found using LittleBrain, a gradient-based tool to aid interpretation of topological neuroimaging findings of the cerebellum (Guell et al., 2019).

LittleBrain creates a two-dimensional representation of all voxels in the cerebellum, with each axis representing one of the principal functional gradients described by Guell and colleagues (Guell et al., 2018). Gradient 1 stretches from primary motor to non-motor areas, such as language and default regions. This is analogous to functional organization principles previously reported in the cerebral cortex that extend from primary unimodal sensory to transmodal regions (Margulies et al., 2016). Gradient 2 can be understood as a characterization of task focus or cognitive load. This gradient ranges from the two extremes of gradient 1 (motor and default regions) to areas involved in focused cognitive processing, such as working memory or attention.

We mapped standardized age coefficient from the anatomical and functional parcellation using the LittleBrain toolbox and found that cerebellar growth follows mostly gradient 2, and might thus be related cognitive demands during development (Figure 4B). Cerebellar regions with small age-related effects are mainly localized in task unfocused regions with low cognitive demand (low gradient 2 values), such as motor processing and default networks. Regions with larger age-effects can be found in task focused regions (high gradient 2 values), likely to overlap with frontoparietal networks (Guell et al., 2018). Volumetric patterns, principally from the anatomical parcellation, show a more diffuse gradient pattern, possibly due to little overlap between functional activity and macroscopic anatomy (Brett et al., 2002).

Figure 4: Visualizations of growth gradients. **A)** Linear fit lines through the standardized age β coefficients of anatomical (vermal and mean of both hemispheres) and functional (volume, GMD, and WMD) ROIs in anterior-to-posterior order. Shaded areas indicate the 95% prediction intervals of the linear fit lines. Asterisks in the legend

indicate significant AP growth coefficients (slopes of linear fit lines). Anatomical location of functional parcellation centroids are indicated by numbers in the first panel and listed in **Supplementary Table 5. B**) Growth gradients visualized along two functional gradients using the LittleBrain tool (Guell et al., 2019). Gradient 1 (y-axis) ranges from motor (negative values) to non-motor areas (positive values); Gradient 2 (x-axis) from low (negative values) to high (positive values) task focus/cognitive load. Each dot in the scatterplot represents a voxel in the cerebellum. The color map (scaled per modality to ease comparisons) shows standardized age β coefficients of the cerebellar parcellation a given voxel belongs to.

Discussion (lines 429-456)

Interestingly, the anterior-posterior growth trends in the cerebellum mirror a previously reported cerebellar functional gradient and maturation patterns found in the cerebrum. In the cerebral cortex, similar patterns of earlier maturation of sensorimotor compared to higher-order cognitive areas can be observed in myelination (Deoni et al., 2015; Elston & Fujita, 2014) and grey matter maturation (Giedd et al., 2015; Gilmore et al., 2012; Gogtay & Thompson, 2010; Tamnes et al., 2017), pointing towards directly related growth trajectories of the cerebellum and the cerebrum. Considering the two principle functional gradients that have been described by Guell and colleagues, our reported growth gradients might not only be reflective of motor vs. non-motor involvement but might also pertain to cognitive load. The first gradient extends from motor to non-motor areas, while the second gradient stretches from areas involved in task focused to task unfocused processing (Guell et al., 2018). The growth gradients in both parcellations and over different modalities (volume, GMD, WMD) map best onto gradient two, although volumetric gradient patterns seem to be more diffuse (Figure 4B). This implies earlier maturation of areas involved in task unfocused cognitive processing, like motor function and default mode networks, and later or prolonged maturation of regions likely to share involvement in working memory processing and frontoparietal networks. In the cerebrum, frontoparietal networks are known to mature later as, for example, sensorimotor networks. However, recently age-dependent maturation patterns paralleling the first, but not the second functional cerebellar gradient have been reported, with default networks reaching maturation last in a very similarly aged cohort (Dong et al., 2021). An alternative explanation for this result could have to do with the interplay between default mode and frontoparietal networks. Recently, the default mode network has been proposed to serve as a compensatory scaffold to support executive functions in children and young adults with immature frontoparietal network (Chen et al., 2023). Importantly, while there are suggestions of the growth patterns resembling gradient 2 proposed by Guell and colleagues, the depiction of the gradients using the LittleBrain toolbox in the current study is inconclusive, mainly due to divergent patterns in the anatomical parcellations. The issue of age related changes in cerebellar functional networks remains to be closely examined and could be explored in future studies using large-scale, longitudinal functional neuroimaging data. Together with similar maturation patterns in myelination and grey matter density, these findings provide additional support for developmental interactions between the cerebellum and cerebrum.

Conclusion (line 566-568)

The anterior/sensorimotor-posterior/cognitive growth gradient resembles a recently proposed functional gradient related to cognitive load and follows cerebral maturation patterns, thus providing evidence for directly related cerebello-cortical developmental trajectories.

19) The data for waves 2 and 3 are heavily enriched in two age bands (wave 1 seems more evenly distributed between 7 and 10 yrs). How might this impact the accuracy of the growth curves for ages where there is relatively little data available?

In general it is true that the estimated normative model will have reduced precision in regions of the age range having small numbers of data points, particularly when such regions are narrow. However, in this study we use an HBR framework which approximates the nonlinear age trajectory with a piecewise linear trajectory, (i.e. linear within every site) and is therefore less susceptible to this problem than more complex nonlinear models. It is also important to note that waves 2 and 3 were acquired on the same scanner, which effectively anchors the trajectory to interpolate these time points, thereby further reducing the uncertainty in the estimated centiles.

20) It seems like a missed opportunity not to examine the subset of individuals with longitudinal data (scans at all 3 timepoints). Do these individuals stay on their growth curves over time?

The question on longitudinal changes is very important and the focus of ongoing research efforts. Generally, differences in normative model deviation scores over time can be due to measurement error or biological changes. To separate these effects, specialized analyses are needed where uncertainty in the centiles and variance at the given time point are estimated. This is currently a very active, yet ongoing, area of research in the field. In a recent preprint, Rehak Buckova et al. propose a pipeline to achieve this and introduce a *z-diff score* to quantitatively describe longitudinal within-subject changes (Rehak Buckova et al., 2023). Given that these methods are still being developed, and that we believe that trying to validate such a technique in our cohort is beyond the scope of the current manuscript, we agree that this is a topic of great interest for the reader, which is why we now mention the approach in the discussion.

We added in manuscript:

Discussion (lines 525-528)

Furthermore, an approach to quantitatively evaluate within-subject changes in longitudinal designs using normative modeling has been proposed recently (Rehak Buckova et al., 2023), which promises great utility for individual-level data as it allows to estimate whether individual participants or patients follow their expected centiles.

Reviewer #3 (Remarks to the Author):

Here the authors examine normative developmental trajectories of the cerebellum and its constituent regions, something that has been lacking within human neuroscience for many years. The authors use a large neuroimaging dataset collected in the Netherlands whose longitudinal nature allows for powerful developmental analyses. They show that subsections of the cerebellum, whether defined anatomically or functionally, show distinct maturational rates, with interesting sex differences as well. Having created normative growth charts for regions of the cerebellum, the authors then go onto show that individuals with autism-related behavioral traits show significant deviation in several regions of the cerebellum, generally showing less cerebellar tissue. The authors provide a nice code-based framework for future researchers to take advantage of this large dataset. Overall these results are novel, statistically rigorous, and will be of interest to the field.

1) Figure 2, it would be helpful here to maybe have a bar graph or something similar showing the actual growth or %-change for each lobule. I like the renderings of the cerebellum, but having more quantitative representation alongside it so the reader can see what each lobule is doing would be useful. I don't think a bar graph would take up too much space and could be fit alongside the volume renderings. I ask because in Fig 2B, it seems as if Crus II and CM are increasing by almost 25% in volume from 7 to 17 years old, that seems like a lot! I think it's generally interpreted in the field that the brain (probably the cortex) has reached about 80% of its volume by the age of 5, so to show the cerebellum (or parts of it) is exceeding this would be something worth highlighting.

We thank the reviewer for highlighting this important point, and agree it deserves more attention in the manuscript. We have incorporated new graphs into the manuscript that show the growth of regions-of-interest (ROIs) in 2 ways. First, we included bar graphs of standardized age coefficients in the main figures 2 and 3. Standardized coefficients allow for effect comparisons across atlases (anatomical vs. functional), modalities (volume vs. GMD vs. WMD), and can also be employed when comparing effects across studies. Second, we included percentage changes of the mean linear trajectories for each ROI in the supplement.

We added in manuscript:

Figure 2 (lines 208-217)

Figure 2: Effect of age on volume in the anatomical parcellation. **A)** Mean posterior distribution for the standardized age β coefficient (slope) for each anatomical ROI and absolute differences in effect sizes (standardized β) between males and females are illustrated. **B)** Trajectories of males (in yellow) and females (in green in 3 example ROIs: left Lobule V (anterior cerebellum), left Crus II (posterior cerebellum), and left corpus medullare (white matter tract). The bold lines represent the mean trajectories, shaded areas represent what is within 2 standard deviations of the mean. Volume is shown in cubic centimeters (ccm). **C)** Bar graphs of all standardized age β coefficients (slopes) of males and females. Error bars depict 95% confidence interval of the mean. Exact numbers can be found in **Supplementary Table 3** and the percentage change of mean trajectories for each anatomical ROI is illustrated in **Supplementary Figure 3A**.

Figure 3 (lines 255-264)

Figure 3: Effect of age on volume in the functional parcellation. **A)** MDTB functional atlas regions. Figure from King et al. 2019. **B)** Mean posterior distribution for the standardized age β coefficient (slope) for each functional ROI of the MDTB atlas (a-f) and absolute differences in effect size (standardized β) between males and females are illustrated (g-i). **C)** Trajectories of males (in yellow) and females (in green) for 2 example ROIs. 1: *Left hand presses* (anterior cerebellum) and 5: *Divided attention (left)* (posterior cerebellum). The bold lines represent the mean trajectories, shaded areas represent what is within 2 standard deviations of the mean. **D)** Bar graphs of all standardized age β coefficients (slopes) of males and females. Error bars depict 95% confidence interval of the mean. Exact numbers can be found in **Supplementary Table 4** and the percentage change of mean trajectories for each functional ROI is illustrated in **Supplementary Figure 3B**.

Supplementary Figure 3 (lines 961-966)

Supplementary Figure 3: Percentage change of the mean linear trajectory of males and females between the ages of 6 and 17 for the A) anatomical and B) functional parcellation. Horizontal lines depict the mean percentage change over ROIs, error bars represent percentage change with +/- 1 standard deviation of the mean trajectory. Important to note: percentage change is highly sensitive to initial values. The more extreme the initial value, the more likely is a high percentage change (e.g.: WMD autobiographical recall where initial values are very low (see Supplementary Figures 2C-D)).

2) Individual images were linearly then nonlinearly aligned to a shared space. It later says that parcellations were done in native space. Can the author’s clarify? After reading the methods section “Anatomical Parcellation” a few times, I think I finally understood what was happening. Five hand-

segmented images are nonlinearly aligned to 21 representative brains spanning the age-range to create a small library of 105 segmented cerebella. Then an individual's brain is aligned to each of these 105 segmentations, and then a compression step happens (through voxel-wise majority voting) to get a final segmentation in an individual's brain. Is that correct? If not then this section might benefit from more explanation.

The reviewer is indeed correct, as this is precisely the anatomical image analysis. However, we agree that this section can benefit from adaptations to improve the clarity.

We changed the following in the manuscript:

Methods, Anatomical parcellation (lines 678-694)

The MAGeTBrain framework uses an automated labeling algorithm based on five manually segmented MR images from healthy participants. Non-linear registration is used to align the five manually segmented images (in MAGeTBrain referred to as "atlases") to a series of individual study-specific "template" images (referred to as "templates" here).

Seven unique and representative images from the three study time points were selected, by a) excluding scans with dental implants, b) pre-selecting the top 20 scans with the highest quality ratings based on automatic FreeSurfer quality assurance (Rosen et al., 2018), and c) thoroughly inspecting the top 20 scans per time point for artifacts, inhomogeneities, full coverage of the cerebellum, and (cerebellar) cysts. Based on this evaluation, the 7 scans with the highest quality ratings were selected per time point, resulting in 21 study-specific template images.

Each of the five manually segmented atlases were then applied to the 21 study-specific templates, resulting in 105 cerebellar atlas-template segmentations. This allows for the manual segmented atlases to be propagated to each of the template images. Next, each individual scan in the dataset was non-linearly registered to each of the 105 cerebellar template segmentations, resulting in 105 segmentations for each input image, and enabling the template-atlas labels to be propagated to each individual participant space (Chakravarty et al., 2013). In the final step, for each individual input image, the 105 co-registered atlas-template labels were then fused using voxel-wise majority voting to create a final segmentation.

3) It would help to have a more intuitive description of what gray/white matter density are. Is it essentially how gray or white a given voxel is? Is it derived from the inherent voxel intensity within the individual's image, or is it derived by aligning the individual to the MNI space and is it extracted from some probability map there? This can be useful for interpretation. For example in Fig 4B it seems that GMD decreases and WMD increases, which might be in line with the interpretation that myelin content is increasing (thus whitening most voxels).

SMRIPrep, the automated tool we used for preprocessing all images, calls the FAST tissue segmentation tool from FSL. FSL FAST outputs tissue-type probability map for each tissue type (here: GMD, WMD, and CSF). That is, given the inherent voxel intensity of a (brain extracted) scan (in the context of the other voxels and distribution of intensities), the probability of a given

voxel being grey matter or white matter (or CSF). We now clarify this in the manuscript and mention the tissue segmentation tool (FSL FAST) specifically.

We added in the manuscript:

Methods, Image pre-processing (lines 639-642)

The tissue segmentation procedure (FSL FAST: <https://fsl.fmrib.ox.ac.uk/fsl/fslwiki/FAST>) resulted not only in binary classifications of voxels, but also in per-voxel tissue class probability estimates. These probability estimates can be interpreted as the likelihood of a given voxel being grey matter, white matter, or cerebrospinal fluid.

4) Figure 5 is nice. However, when ranking functional parcels onto the AP axis, the authors write that the volumetric/spatial centroid was used to determine in which anatomical lobule a given functional parcel falls. As the authors note, most functional parcels have two components (left-hand presses show representation in anterior and posterior lobules) and if the centroid falls outside of a lobule they hand-assign it to the nearest lobule. For the case of left-hand presses, that might be a lobule that doesn't actually have any left-hand press representation. Does that happen? The authors acknowledge that this approach isn't perfect. Maybe they could also just try a binary test comparing motor versus non-motor representations to test for gradient difference in development? Related to this point, Guell and colleagues in their "functional gradients of the cerebellum" article find an A-P gradient (gradient 1) from functional connectivity, maybe using those values to assign your functional parcels a better sensory-cognitive score (rather than a spatial centroid) would help? The sex difference flatmaps in Figure 4A for GMD and WMD seem to map onto the Guell gradient 1 map quite well (peaks near Crus1/2, low values in anterior lobules like Lob1-III and IX) so it could be a worthwhile endeavor. Also, and perhaps this is just a semantic difference, but isn't lobule IX as anterior as lobule I-III? Isn't Crus I/II the most posterior?

Thank you for the comment. In the following, we will answer the points concerning centroid locations, gradient difference in motor vs. non-motor parcels, and anterior/posterior labelling of lobules separately.

- The centroid for the left-hand presses falls within the left lobule 5 which does contain functional representations of left-hand presses. However, the centroid of the right-hand presses falls outside the right lobule 5 by just one or two voxels. Please see attached image for illustration. The same is true for the only other centroid (parcel 5: divided attention (left) – centroid just outside of Crus I) that did not fall inside lobular boundaries.

Centroid Locations

- Thank you for the suggestion of using the functional gradients proposed by Guell as a comparison (Guell et al., 2018). We added an analysis using the LittleBrain gradient toolbox (Guell et al., 2019) – please refer to comment 18 of reviewer 2.
- Concerning the description of lobules as anterior or posterior, this can indeed be misleading as there is a difference between what is anterior/posterior in the cerebrum compared to the what is commonly referred to as anterior/posterior in the cerebellum. Lobules III – V together make up the anterior lobe of the cerebellum (lobules I and II are located in the midline [anterior vermis]). The primary fissure separates the anterior from the posterior lobe (lobules VI-IX), which is why lobules IX are referred to as posterior while lobule III as anterior. Lobule III is therefore the most anterior, and lobule IX the most posterior lobule of the cerebellar hemispheres, despite the fact that they are both located in the same (or adjacent) coronal plane.

5) Within the field of neuroimaging, the volumetric findings from autism cohorts on the brain and cerebellum are a little all over the place, so in some sense the authors' findings using this massive dataset (the largest study of the cerebellum to date) stand to add a lot of clarity. I think the authors could increase their discussion on this point, pointing out which studies they replicate, and perhaps discussing that their parcellated approach (rather than treating the cerebellum as a single structure) perhaps allowed for better sensitivity.

Thank you, we agree that our large scale approach will allow for better characterizations of disease patterns in autism but also in other pathologies. Importantly, we use the example of children with high social responsiveness score (SRS) in our general cohort to illustrate the utility of the approach. We expanded our discussion on previous findings and highlight the parcellated approach.

We added in the manuscript:

Discussion (lines 477-495)

In accordance with previous research, we find smaller cerebellar volumes in children with autistic traits. In the anatomical parcellation smaller volumes can be seen throughout various

regions, particularly in vermal and lobular parts of the anterior and superior posterior cerebellum (Figure 5A). *This corroborates findings of hypoplasia in posterior vermal and lobular regions which have been reported consistently before in clinical samples (D'Mello et al., 2015; Kaufmann et al., 2003; McKinney et al., 2022; Pierce & Courchesne, 2001; Stanfield et al., 2008).* While the functional parcellation also reveals smaller volumes throughout almost all ROIs, with significant differences in MDTB components 1) Left-hand presses and 6) Divided attention (right), the MDTB components 7) Narrative and 8) Word comprehension seem to not follow the same trend (Figure 5B). Given the overlap of MDTB components 7 and 8 with the cerebellar default mode regions described by Buckner and colleagues (Buckner et al., 2011), a network found to be among the most disrupted in ASD patients, this might relate to previously reported heterogeneity in default mode network connectivity in children on the Autism spectrum (Padmanabhan et al., 2017). Clear differences can also be observed in GMD with a high percentage of individuals with autistic traits exhibiting increased GMD in the anterior sensorimotor parcels, particularly on the left hemisphere, and decreased GMD in the superior posterior parcels involved in language processing (Figure 5C). In view of the heterogeneity in brain morphology between individuals across a multitude of pathologies, it is noteworthy that the current approach does not depend on group-level inferences but can be used at an individual level to uncover within-group heterogeneity. *Furthermore, the availability of a very detailed anatomical as well as a functional parcellation allow for more sensitive approaches when investigating pathological deviations from typical development.*

Minor Comments:

6) CCM is used as the unit for volume analyses (e.g., Figure 2B). What does it stand for? Is it cubic centimeters? I'm not sure the acronym is defined anywhere.

This is correct. We now define ccm as cubic centimeters in the figure legends.

We added in the manuscript:

Figure 2, Legend (line 214)

Volume is shown in cubic centimeters (ccm).

Supplementary Figure 2A&B, Legend (lines 933-934 & 941-942)

The y-axis shows volume in cubic centimeters (ccm).

7) Seems like Fig 3 could be added as an inset into Fig 4 given how related they are (and that 3 is just a reproduction).

Thank you for the comment. To have a better overview, the reproduction of the functional parcellations (King et al., 2019) is now included in Figure 3 (before: Figure 4). Please see the response to comment 1.

Reviewer #4 (Remarks to the Author):

1. What are the noteworthy results?

Comment: The first normative model of anatomical and functional subregions of the cerebellum was established from a large pediatric population, and revealed an anterior-posterior gradient of human cerebellum.

2. Will the work be of significance to the field and related fields? How does it compare to the established literature? If the work is not original, please provide relevant references.

Comment: The present work is highly novel and holds great significance to the field of human brain mapping as well as related fields such as brain disorders and public health. Some novel results (e.g., the A-P gradient of brain growth is highly implicated with previous literature on brain maturation such as cortical gradient development).

3. Does the work support the conclusions and claims, or is additional evidence needed?

Comment: Overall, the work has been done in a very solid way. But, it would be more informative if the authors can demonstrate some valid usage of the proposed normative models, for example, the association studies on the cerebellum and behavior/cognition.

Thank you for your comments on points 1 and 2.

Besides illustrating how the models can be used in a clinical phenotype (autistic traits), we additionally added a supplementary analysis on associations between cerebellar functional regions of interest and IQ. Please refer to comment 6 of reviewer 1 for full details.

4. Are there any flaws in the data analysis, interpretation and conclusions? Do these prohibit publication or require revision? Is the methodology sound? Does the work meet the expected standards in your field?

Comment: A problematic point is about the unequal distribution of sample ages, which could be the driving force of the overall linear trajectories modeled. The authors need to be careful for the interpretation on their findings while they can do some validation based on some smaller-scale cohorts such as Healthy Brain Network or Chinese Color Nest. These cohorts are openly shared to the community.

Thank you for raising the point of an external validation. We fully agree that generalizability is an important issue in modeling approaches, which is why we used a 50% train / 50% test split in our cohort, and therefore utilizing half of the data for validation on a holdout set. The computational time and the resources needed to run the anatomical parcellation (MAGeTBrain) in addition to the ensuing quality control, would not allow us to include a new cohort within a short timeframe. Furthermore, a recent paper on comparisons of normative models suggest that model parameters become highly stable with $N > 3,000$ (Ge et al., 2023). Nevertheless, we agree with the reviewer on the importance of external validation steps, and therefore included this suggestion in the limitation section of the discussion.

Concerning the effect of having peaked distributions in 2 measurement waves of the model estimates, we kindly refer to comment 19 of reviewer 2.

We added in the manuscript:

Discussion, Limitations (lines 540-547)

While normative models were found to be highly stable with $N > 3,000$ (Ge et al., 2023), it is noteworthy that the current models can easily be updated within the PCNtoolkit framework. This can also include new data points outside of our age range and from diverse backgrounds, or from clinical cohorts. As a consequence, the cerebellar normative models can be extended and refined as new information becomes available while new, possibly smaller cohorts can benefit from informed priors based on our models. In such a way, the current models and our results on deviations in children likely to fall on the autism spectrum can be validated in an external (clinical) cohort in the future.

5. Is there enough detail provided in the methods for the work to be reproduced?

Comment: Yes.

We are happy the reviewer agrees our methods are comprehensively presented.

6. Other Comments:

The scientific finding of the A-P growth gradient is very interesting while can be further informed with recent advances on developmental shifts of cortical connectivity gradient from childhood to adolescence, please refer to <https://pubmed.ncbi.nlm.nih.gov/34260385>.

Thank you for this valuable addition. We now compare results of the suggested paper with the growth gradients reported in the current manuscript in the discussion section.

We added in the manuscript:

Discussion (lines 442-454)

In the cerebrum, frontoparietal networks are known to mature later as, for example, sensorimotor networks. However, recently age-dependent maturation patterns paralleling the first, but not the second functional cerebellar gradient have been reported, with default networks reaching maturation last in a very similarly aged cohort (Dong et al., 2021). An alternative explanation for this result could have to do with the interplay between default mode and frontoparietal networks. Recently, the default mode network has been proposed to serve as a compensatory scaffold to support executive functions in children and young adults with *immature* frontoparietal network (Chen et al., 2023). Importantly, while there are suggestions of the growth patterns resembling gradient 2 proposed by Guell and colleagues, the depiction of the gradients using the LittleBrain toolbox in the current study is inconclusive, mainly due to

divergent patterns in the anatomical parcellations. The issue of age related changes in cerebellar functional networks remains to be closely examined and could be explored in future studies using large-scale, longitudinal functional neuroimaging data.

References

- Chakravarty, M. M., Steadman, P., Van Eede, M. C., Calcott, R. D., Gu, V., Shaw, P., Raznahan, A., Collins, D. L., & Lerch, J. P. (2013). Performing label-fusion-based segmentation using multiple automatically generated templates. *Human brain mapping, 34*(10), 2635-2654.
- Chen, M., He, Y., Hao, L., Xu, J., Tian, T., Peng, S., Zhao, G., Lu, J., Zhao, Y., & Zhao, H. (2023). Default mode network scaffolds immature frontoparietal network in cognitive development. *Cerebral cortex, 33*(9), 5251-5263.
- Deoni, S. C., Dean, D. C., 3rd, Remer, J., Dirks, H., & O'Muircheartaigh, J. (2015). Cortical maturation and myelination in healthy toddlers and young children. *Neuroimage, 115*, 147-161.
<https://www.ncbi.nlm.nih.gov/pubmed/25944614>
- Diedrichsen, J. (2006). A spatially unbiased atlas template of the human cerebellum. *Neuroimage, 33*(1), 127-138.
- Dong, H.-M., Margulies, D. S., Zuo, X.-N., & Holmes, A. J. (2021). Shifting gradients of macroscale cortical organization mark the transition from childhood to adolescence. *Proceedings of the National Academy of Sciences, 118*(28), e2024448118.
- Elston, G. N., & Fujita, I. (2014). Pyramidal cell development: postnatal spinogenesis, dendritic growth, axon growth, and electrophysiology. *Front Neuroanat, 8*, 78.
<https://doi.org/10.3389/fnana.2014.00078>
- Ge, R., Yu, Y., Qi, Y. X., Fan, Y. V., Chen, S., Gao, C., Haas, S. S., Modabbernia, A., New, F., & Agartz, I. (2023). Normative Modeling of Brain Morphometry Across the Lifespan using CentileBrain: Algorithm Benchmarking and Model Optimization. *bioRxiv, 2023.2001.2030.523509*.
- Giedd, J. N., Raznahan, A., Alexander-Bloch, A., Schmitt, E., Gogtay, N., & Rapoport, J. L. (2015). Child psychiatry branch of the National Institute of Mental Health longitudinal structural magnetic resonance imaging study of human brain development. *Neuropsychopharmacology, 40*(1), 43-49.
- Gilmore, J. H., Shi, F., Woolson, S. L., Knickmeyer, R. C., Short, S. J., Lin, W., Zhu, H., Hamer, R. M., Styner, M., & Shen, D. (2012). Longitudinal development of cortical and subcortical gray matter from birth to 2 years. *Cereb Cortex, 22*(11), 2478-2485.
<https://www.ncbi.nlm.nih.gov/pubmed/22109543>
- Gogtay, N., & Thompson, P. M. (2010). Mapping gray matter development: implications for typical development and vulnerability to psychopathology. *Brain Cogn, 72*(1), 6-15.
<https://www.ncbi.nlm.nih.gov/pubmed/19796863>
- Guell, X., Goncalves, M., Kaczmarzyk, J. R., Gabrieli, J. D. E., Schmahmann, J. D., & Ghosh, S. S. (2019). LittleBrain: a gradient-based tool for the topographical interpretation of cerebellar neuroimaging findings. *Plos one, 14*(1), e0210028.
- Guell, X., Schmahmann, J. D., Gabrieli, J. D. E., & Ghosh, S. S. (2018). Functional gradients of the cerebellum. *Elife, 7*, e36652.
- King, M., Hernandez-Castillo, C. R., Poldrack, R. A., Ivry, R. B., & Diedrichsen, J. (2019). Functional boundaries in the human cerebellum revealed by a multi-domain task battery. *Nat Neurosci, 22*(8), 1371-1378. <https://www.ncbi.nlm.nih.gov/pubmed/31285616>
- Marek, S., Tervo-Clemmens, B., Calabro, F. J., Montez, D. F., Kay, B. P., Hatoum, A. S., Donohue, M. R., Foran, W., Miller, R. L., & Hendrickson, T. J. (2022). Reproducible brain-wide association studies require thousands of individuals. *Nature, 603*(7902), 654-660.

- Marques, J. P., Van Der Zwaag, W., Granziera, C., Krueger, G., & Gruetter, R. (2010). Cerebellar cortical layers: in vivo visualization with structural high-field-strength MR imaging. *Radiology*, 254(ARTICLE), 942-948.
- Park, M. T., Pipitone, J., Baer, L. H., Winterburn, J. L., Shah, Y., Chavez, S., Schira, M. M., Lobough, N. J., Lerch, J. P., Voineskos, A. N., & Chakravarty, M. M. (2014). Derivation of high-resolution MRI atlases of the human cerebellum at 3T and segmentation using multiple automatically generated templates. *Neuroimage*, 95, 217-231.
<https://www.ncbi.nlm.nih.gov/pubmed/24657354>
- Rehak Buckova, B., Frazza, C., Rehak, R., Kolenic, M., Beckmann, C., Spaniel, F., Marquand, A., & Hlinka, J. (2023). Using normative models pre-trained on cross-sectional data to evaluate longitudinal changes in neuroimaging data. *bioRxiv*, 2023.2006. 2009.544217.
- Rosen, A. F. G., Roalf, D. R., Ruparel, K., Blake, J., Seelaus, K., Villa, L. P., Ciric, R., Cook, P. A., Davatzikos, C., & Elliott, M. A. (2018). Quantitative assessment of structural image quality. *Neuroimage*, 169, 407-418.
- Shaw, P., Ishii-Takahashi, A., Park, M. T., Devenyi, G. A., Zibman, C., Kasperek, S., Sudre, G., MangalMurti, A., Hoogman, M., & Tiemeier, H. (2018). A multicohort, longitudinal study of cerebellar development in attention deficit hyperactivity disorder. *Journal of Child Psychology and Psychiatry*, 59(10), 1114-1123.
- Tamnes, C. K., Herting, M. M., Goddings, A.-L., Meuwese, R., Blakemore, S.-J., Dahl, R. E., Güroğlu, B., Raznahan, A., Sowell, E. R., & Crone, E. A. (2017). Development of the cerebral cortex across adolescence: a multisample study of inter-related longitudinal changes in cortical volume, surface area, and thickness. *Journal of Neuroscience*, 37(12), 3402-3412.
- Tiemeier, H., Lenroot, R. K., Greenstein, D. K., Tran, L., Pierson, R., & Giedd, J. N. (2010). Cerebellum development during childhood and adolescence: a longitudinal morphometric MRI study. *Neuroimage*, 49(1), 63-70.

Reviewer #1 (Remarks to the Author):

The authors have answered in great details to my remarks. In general, I think that the quality of the manuscript have improved.

I would still have a question regarding the interpretation of the new data presented in supplementary material 8.

For the volume (left panel), I think that the results are coherent. In the low IQ vs high IQ group, there seems to be an opposite pattern. However for GMD (middle panel), there are regions of interest in the cognitive right cerebellum where the pattern is identical in the low vs high IQ group. Does that mean the children with low and high IQ have higher GMD than expected in the same ROIs ? I think that it would be interesting if the authors could elaborate on these results.

Reviewer #2 (Remarks to the Author):

The authors have adequately addressed most of my questions and comments. The new analyses address the open questions that I had and more solidly ground this study in the context of the existing literature. The description of the methods is greatly improved and much clearer.

I only have minor comments at this stage:

- I recommend that the authors incorporate the information from the legend for Supplementary Fig 7 into the main text: "The MAGeT algorithm subdivides the cerebellum into 11 vermal and 22 hemispheric lobules (11 on each hemisphere). Additionally, the central white matter, the corpus medullare, is segmented in each hemisphere. White matter that extends into the folia of the lobules was segmented as part of the lobules." This way the information is readily available in the main Methods section. This is important given that different parcellation methods deal with cerebellar lobules (particularly I-II, III) in different ways.
- Figs 2 and 3 – the error bars are impossible to see, can this be remedied?
- I think it is important that the authors continue to emphasize throughout the manuscript that the age range they are studying (6+ years) is one in which you would expect the most developmental changes in association / not sensorimotor regions. In other words, the findings don't mean that there are not age-related changes in the core motor regions of the cerebellum, just that they are not captured in this particular age range. It would be helpful to add "in ages 6-17 years" or "in the age range measured here" or "during childhood and adolescence" similar statements for clarity and to capture this nuance throughout the manuscript.
- P. 14, line 306-308 "We mapped standardized age coefficient from the anatomical and functional parcellation using the LittleBrain toolbox and found that cerebellar growth follows mostly gradient 2, and might thus be related to cognitive demands during development (Figure 4B)." Another interpretation might be that the growth is not necessarily related to cognitive demands changing (which suggests the direction is cognitive demands  driving the growth) and could instead be related to the typical maturation patterns that are seen in the cerebral cortex, where the development of areas associated with higher-level cognitive processes emerge later. The authors do note this elsewhere in the manuscript. A minor edit might be something like "... cerebellar growth follows mostly gradient 2 and might thus be related to the documented later maturation of brain regions supporting higher-level cognitive processes".
- P. 16, line 348 "... in vermal and lobular regions". Recommended change to "... in vermal and hemisphere regions"
- P. 19, lines 413-415. There are functional differences throughout the vermis, e.g. the oculomotor vermis, and the posterior vermis is more associated with emotional regulation / processing than the anterior vermis. The authors might want to rephrase this sentence to reflect those nuances.
- P. 20. Another caveat in terms of the anterior-posterior gradient is that there is more than one representation of the functional connectivity networks (e.g. motor to cognitive networks, then cognitive to motor, with a shift in lobule VII – see Buckner et al. 2011), so it may not be linear as you move from lobules I-X

Reviewer #3 (Remarks to the Author):

The authors have performed a number of analyses and textual changes that have significantly

improved the manuscript and addressed any minor concerns I may have add. I am satisfied with the manuscript in its new state and congratulate the authors on a nice collection of findings.

REVIEWERS' COMMENTS

Reviewer #1 (Remarks to the Author):

The authors have answered in great details to my remarks. In general, I think that the quality of the manuscript have improved.

I would still have a question regarding the interpretation of the new data presented in supplementary material 8.

For the volume (left panel), I think that the results are coherent. In the low IQ vs high IQ group, there seems to be an opposite pattern. However for GMD (middle panel), there are regions of interest in the cognitive right cerebellum where the pattern is identical in the low vs high IQ group. Does that mean the children with low and high IQ have higher GMD than expected in the same ROIs ? I think that it would be interesting if the authors could elaborate on these results.

We thank the reviewer and have addressed the last remaining point in the supplement due to space constraints in the main text. We indeed show somewhat similar trends in GMD and WMD in children with high and low IQ. It is important to note that IQ might not relate linearly to brain structure. Using the current approach, future studies could investigate how such patterns might differ on an individual level, or how comorbidities might influence such relationships i.e. high IQ with or without ASD.

We added in the supplement:

Supplementary information, Supplementary Figure 8 (lines 96-98)

Interestingly, high deviations in the same ROIs can be seen in the low and high IQ group in GMD and WMD. This might relate to non-linear effects of IQ on brain structure. Source data are provided as a Source Data file.

Reviewer #2 (Remarks to the Author):

The authors have adequately addressed most of my questions and comments. The new analyses address the open questions that I had and more solidly ground this study in the context of the existing literature. The description of the methods is greatly improved and much clearer.

I only have minor comments at this stage:

- I recommend that the authors incorporate the information from the legend for Supplementary Fig 7 into the main text: "The MAGeT algorithm subdivides the cerebellum into 11 vermal and 22 hemispheric

lobules (11 on each hemisphere). Additionally, the central white matter, the corpus medullare, is segmented in each hemisphere. White matter that extends into the folia of the lobules was segmented as part of the lobules.” This way the information is readily available in the main Methods section. This is important given that different parcellation methods deal with cerebellar lobules (particularly I-II, III) in different ways.

We thank the reviewer for his/her time and valuable comments. Regarding the first point, we have added the information in the methods section.

We added in the manuscript:

Methods, Anatomical parcellation (lines 651-656)

The MAGeT algorithm subdivides the cerebellum into 11 vermal and 22 hemispheric lobules (11 on each hemisphere). Additionally, the central white matter, the corpus medullare, is segmented in each hemisphere. White matter that extends into the folia of the lobules was segmented as part of the lobules. Volumes for each of these 35 anatomical parcellations are generated in mm³ by the MAGeT pipeline. Supplementary Figure 7 shows a representative automatically labelled segmentation from one individual and gives an overview of all 35 anatomical subdivisions.

- Figs 2 and 3 – the error bars are impossible to see, can this be remedied?

We agree and chose to show the standard deviation instead of the 95% confidence interval of the mean in the updated figures 2 and 4 (formerly figure 3):

Figure 2: Effect of age on volume in the anatomical parcellation.

A) Mean posterior distribution for the standardized age β coefficient (slope) for each anatomical ROI and absolute differences in effect sizes (standardized β) between males and females are illustrated. B) Trajectories of males (in yellow) and females (in green) in 3 example ROIs: left Lobule V (anterior cerebellum), left Crus II (posterior cerebellum), and left corpus medullare (white matter tract). The bold lines represent the mean trajectories, shaded areas represent what is within 2 standard deviations of the mean. Volume is shown in cubic centimeters (ccm). C) Bar graphs of all standardized age β coefficients (slopes) of males (in yellow) and females (in green). Error bars depict ± 1 standard deviation of standardized age β samples ($n=12,000$). Exact numbers can be found in Supplementary Table 3 and the percentage change of mean trajectories for each anatomical ROI is illustrated in Supplementary Figure 3A. Source data are provided as a Source Data file.

Figure 4: Effect of age on volume, *Grey Matter Density* (GMD), and *White Matter Density* (WMD) in the functional parcellation. A) Mean posterior distribution for the standardized age β coefficient (slope) for each functional ROI of the MDTB atlas (a-f) and absolute differences in effect size (standardized β) between males (in yellow) and females (in green) are illustrated (g-i). B) Trajectories of males (in yellow) and females (in green) for 2 example ROIs. 1: Left hand presses (anterior cerebellum) and 5: Divided attention (left) (posterior cerebellum). The bold lines represent the mean trajectories, shaded areas represent what is within 2 standard deviations of the mean. C) Bar graphs of all standardized age β coefficients (slopes) of males and females. Error bars depict ± 1 standard deviation of standardized age β samples (n=12,000). Exact numbers can be found in Supplementary Table 4 and the percentage change of mean trajectories for each functional ROI is illustrated in Supplementary Figure 3B. Source data are provided as a Source Data file.

- I think it is important that the authors continue to emphasize throughout the manuscript that the age range they are studying (6+ years) is one in which you would expect the most developmental changes in association / not sensorimotor regions. In other words, the findings don't mean that there are not age-related changes in the core motor regions of the cerebellum, just that they are not captured in this particular age range. It would be helpful to add "in ages 6-17 years" or "in the age range measured here" or "during childhood and adolescence" similar statements for clarity and to capture this nuance throughout the manuscript.

We adjusted the manuscript as follows:

Results, Functional parcellation (lines 220-225)

While it is well-documented that GMD decreases and WMD increases in the brain during this age range, we again see a clear distinction between anterior motor regions and posterior cognitive regions in 6 to 17 year olds. This is further illustrated by the growth trajectories of an example anterior motor (1: Left hand presses) and posterior cognitive (5: Divided attention (left)) ROI. Steeper slopes, and thus more developmental changes, are observed in posterior cognitive regions compared to anterior motor regions during childhood and adolescence (Figure 4B a, b & c).

Results, Anterior-posterior growth gradient (lines 274-276)

We mapped standardized age coefficient from the anatomical and functional parcellation using the LittleBrain toolbox and found that cerebellar growth during childhood and adolescence follows mostly gradient 2, and might thus be related to cognitive demands during development (Figure 5B).

Discussion (lines 353-356)

Between ages 6 and 17, anterior sensorimotor areas show smaller age-related effects compared to posterior cognitive areas, possibly reflecting protracted growth trajectories for higher-order cognitive compared to sensorimotor regions in the cerebellum.

- P. 14, line 306-308 "We mapped standardized age coefficient from the anatomical and functional parcellation using the LittleBrain toolbox and found that cerebellar growth follows mostly gradient 2, and might thus be related to cognitive demands during development (Figure 4B)." Another interpretation might be that the growth is not necessarily related to cognitive demands changing (which suggests the direction is cognitive demands  driving the growth) and could instead be related to the typical maturation patterns that are seen in the cerebral cortex, where the development of areas associated with higher-level cognitive processes emerge later. The authors do note this elsewhere in the manuscript. A minor edit might be something like "... cerebellar growth follows mostly gradient 2 and might thus be related to the documented later maturation of brain regions supporting higher-level cognitive processes".

Thank you for raising this point. We do agree and that the growth changes seem to follow age-related improvements of underlying function (higher-level functions in late childhood and adolescence) and cover this point in the discussion. However, we politely disagree that the LittleBrain tool is an appropriate approach to prove this assumption. Both gradients in the LittleBrain toolbox (Gradient 1: non-motor to motor areas; Gradient 2: low to high task focus/cognitive load) could in theory support the later maturation of high-level area theory. Therefore, the LittleBrain toolbox, if a pattern can be observed, would always support such a trend but is not able to prove the absence of it.

- P. 16, line 348 "... in verbal and lobular regions". Recommended change to "... in vermal and hemisphere regions"

We changed the manuscript accordingly:

Results, Large normative model deviations and clinical or behavioral phenotypes (lines 302-306)

In the anatomical parcellation, a higher percentage of participants with high SRS scores presented with large negative z-scores (smaller volume than expected) throughout various ROIs (Figure 6A), specifically in vermal and hemispheric regions of the anterior and superior posterior cerebellum (significant percentage with large deviations binomial test at $p < 0.05$: Crus I (left), VIIIB (left), vermal region III, Lobule VI (right)).

- P. 19, lines 413-415. There are functional differences throughout the vermis, e.g. the oculomotor vermis, and the posterior vermis is more associated with emotional regulation / processing than the anterior vermis. The authors might want to rephrase this sentence to reflect those nuances.

Thank you for the comment. We adjusted the manuscript to include the information on localized differences in function in the cerebellar vermis.

We changed in the manuscript:

Discussion (lines 365-370)

The possible absence of an anterior-posterior growth gradient in vermal areas is interesting, since functional differences are also present throughout the vermis. However, these are functionally distinct from the hemispheres of the cerebellum as the vermis receives sensorimotor afferents from the spinal cord, and is predominantly involved in lower-order functions, such as postural control, locomotion, and gaze [31], but also plays an important role in emotion processing [32-34].

- P. 20. Another caveat in terms of the anterior-posterior gradient is that there is more than one representation of the functional connectivity networks (e.g. motor to cognitive networks, then cognitive to motor, with a shift in lobule VII – see Buckner et al. 2011), so it may not be linear as you move from lobules I-X

Thank you for the suggestion. We agree and added this information to the caveats of the anterior-posterior gradient discussion.

We added in the manuscript:

Discussion (lines 382-383)

However, results should be interpreted with caution given the complex geometry of the cerebellum and the possibility of non-linear functional gradients (Buckner et al., 2011).

Reviewer #3 (Remarks to the Author):

The authors have performed a number of analyses and textual changes that have significantly improved the manuscript and addressed any minor concerns I may have add. I am satisfied with the manuscript in its new state and congratulate the authors on a nice collection of findings.

We thank the reviewer for his/her time and feedback.